# Delineating organizational principles of the endogenous L-A virus by cryo-EM and computational analysis of native cell extracts

Lisa Schmidt[1,2,3,7], Christian Tüting [1,2,7] ✉, Fotis L. Kyrilis [1,2,4], Farzad Hamdi [1,2], Dmitry A. Semchonok [1], Gerd Hause[5], Annette Meister[1,2], Christian Ihling[6], Milton T. Stubbs [1,2], Andrea Sinz [6] & Panagiotis L. Kastritis [1,2,4,5] ✉

The high abundance of most viruses in infected host cells benefits their structural characterization. However, endogenous viruses are present in low copy numbers and are therefore challenging to investigate. Here, we retrieve cell extracts enriched with an endogenous virus, the yeast L-A virus. The determined cryo-EM structure discloses capsid-stabilizing cation-π stacking, widespread across viruses and within the *Totiviridae*, and an interplay of non-covalent interactions from ten distinct capsomere interfaces. The capsid-embedded mRNA decapping active site trench is supported by a constricting movement of two flexible opposite-facing loops. tRNA-loaded polysomes and other biomacromolecules, presumably mRNA, are found in virus proximity within the cell extract. Mature viruses participate in larger viral communities resembling their rare in-cell equivalents in terms of size, composition, and inter-virus distances. Our results collectively describe a 3D-architecture of a viral milieu, opening the door to cell-extract-based high-resolution structural virology.

Viruses typically consist of genetic material enveloped by a protein or a proteolipid shell[1], promoting the transport of their genome from cell to cell. To proliferate, a virus depends on the host cell. A typical viral life cycle is generally divided into 6 steps[2]: attachment, penetration, uncoating, gene expression and replication, viral assembly, and, finally, release of progeny viruses, where the whole process starts anew. Viruses form viral factories during infections by hijacking the host cells' replication machinery or even completely reprograming the host cell[3]. While many viruses inevitably lead to the host cell's damage or death[4,5], detrimental effects on the host's health[6] of endogenous viruses and retroviruses can sometimes be offset by neutral or even beneficial outcomes for their host[7], e.g., shaping innate immune responses[6].

The structural analysis of viruses plays a pivotal role in enhancing our understanding of their epidemiology, ecology, and evolution, as it provides insights into their transfection pathways and interaction targets. Virus diagnostics, traditionally encompassing serological and genetic methods as well as morphological characterizations using various microscopy techniques, including electron microscopy[8] and light microscopy[9], contribute to the detection, evaluation, and handling of viruses in cells. Viral structures at high resolution are studied with X-ray diffraction (crystallography and fiber diffraction) and cryogenic electron microscopy (single-particle cryo-electron microscopy (cryo-EM) or cryo-electron tomography (cryo-ET)) methods.

Current advances in structural biology, especially in cryogenic electron microscopy[10], have allowed an unprecedented analysis of viral architecture across methods and scales (in silico[11], in vitro[12], in situ[13], and in cellulo[14]). Most structural studies focus on exogenous viruses, not only because of their critical importance for understanding the molecular mechanisms of infection but also due to their intrinsic high abundance in the host cell that results

¹Interdisciplinary Research Center HALOmem, Charles Tanford Protein Center, Martin Luther University Halle-Wittenberg, Kurt-Mothes-Straße 3a, Halle/Saale, Germany. ²Institute of Biochemistry and Biotechnology, Martin Luther University Halle-Wittenberg, Kurt-Mothes-Straße 3, Halle/Saale, Germany. ³Technical Biogeochemistry, Helmholtz Centre for Environmental Research, Permoserstraße 15, Leipzig, Germany. ⁴Institute of Chemical Biology, National Hellenic Research Foundation, Athens, Greece. ⁵Biozentrum, Martin Luther University Halle-Wittenberg, Weinbergweg 22, Halle/Saale, Germany. ⁶Institute of Pharmacy, Center for Structural Mass Spectrometry, Martin Luther University Halle-Wittenberg, Kurt-Mothes-Str. 3, Halle (Saale), Germany. ⁷These authors contributed equally: Lisa Schmidt, Christian Tüting. ✉e-mail: christian.tueting@bct.uni-halle.de; panagiotis.kastritis@bct.uni-halle.de

in lysis (e.g., for SARS-CoV-2, an overall yield of ~$10^5$ to $10^6$ virions has been observed per infected cell[15]). On the other hand, endogenous viruses that can integrate into the genome or be present within the host cells in low copy numbers are more challenging to study structurally[16], especially within cells. Consequently, the state-of-the-art in the study of endogenous viruses, as illustrated by recent work of the human endogenous retrovirus K17[17], frequently relies on the overexpression of viral proteins in either heterologous systems or cell cultures. This approach introduces an environment that may affect the native structural properties of the viral particle.

*S. cerevisiae*, a genetically well-described system, harbors in its genome an endogenous virus known as L-A virus[18], a member of the Totivirus family, and its discovery was the main starting point of research on yeast virology[19]. The L-A virus is stably maintained in yeast cells and propagated vertically during mitosis or horizontally through cytoplasmic mixing during mating[20] and appears to be symptomless for yeasts[21]. However, coinfections with satellite viruses—the M1, M2, M28, or Mlus[22] which encode a toxic protein product—lead to the formation of an L-A helper/killer virus system[23]. The two viruses synergize with their host yeast for a killer phenotype, preventing contamination of the strain with other strains that lack the virus pair[23]. In addition, another Totivirus, the L-BC virus, is often associated with an L-A virus infection[24]. The L-BC virus is closely related to the L-A virus, with similar genomic size and similar life cycle, including capsid formation and polymerase activity, but without showing helper activity for the satellite viruses[25].

The L-A virus contains a single 4.6 kb double-stranded RNA (dsRNA)[18], encoding for a capsid protein (Gag), and the viral RNA-dependent RNA polymerase fusion protein (Gag/Pol)[26]. Due to ribosomal frameshifting[27], the predicted Gag:Gag/Pol ratio is produced in 60:1[28,29]. The life cycle of the L-A virus takes place in the cytoplasm of the yeast cell: First, the plus strand is synthesized and transported outside the capsid. Then in the cytoplasm, ribosomes translate the RNA, and Gag, as well as Gag/Pol fusion proteins, will be formed. Gag will form an icosahedral capsid of 60 asymmetric dimers that include two Gag/Pol fusion protein copies, bound to the dsRNA[30]. Lastly, the minus-strand synthesis takes place, and the process begins anew (Fig. 1a, adapted from ref. 31). Double-stranded RNA (dsRNA) viruses differ from other groups of viruses in that their genome is never released from the capsid so that the latter performs not only a protective role but is also part of the mRNA synthesis and modification machinery[32]. Low-resolution single-particle cryo-EM analysis[33] and a crystallographic structure of the capsid at 3.5 Å[34] communicated more than 20 years ago demonstrated similarities of the capsid protein to *Reoviridae*[35]. These structures not only demonstrated that L-A virus Gag violates the tenets of quasi-equivalence by having 120 Gag copies[33,34], but localized the first decapping enzyme described in the literature[36]. Recent studies have shown that the decapping site in both the L-A virus and the L-BC virus is able not only to uncap endogenous mRNAs for degradation decoy generation[37], but also performs a 'cap-snatching' function[38,39].

In this work, we utilize the L-A helper virus as a model system for successful isolation while maintaining endogenous interactions. The advantage of this approach is a closer-to-native observation of the endogenous virus because cell extracts are (a) less complex as compared to a whole-cell lysate or a cell but still retain principles of cellular organization[40–42]; (b) easily accessible to biochemical, biophysical, biocomputational and structural methods that can be combined with modern artificial-intelligence-based methods[43]; and (c) amenable to a selective increase in protein concentrations as a traditional biochemical specimen[40]. Our results collectively characterize the 3D 'architecture' of the endogenous L-A virus milieu, opening the door to structural virology in native cell extracts.

## Results
### Identification of the L-A virus in eukaryotic cell extracts through cryo-EM

To isolate complexes of similar molecular weight within the native cell extract, we performed size exclusion chromatography (SEC) and reproducibly collected the resulting fractions in less than 8 h from harvesting yeast cells (Fig. 1b, Supplementary Fig. 1a, "Methods"). The method, adapted from fractionation experiments on the thermophilic mold *Chaetomium thermophilum*[41] with minor modifications involving changes in starting volume of cells and the process of resuspending the starting material before lysis ("Methods"), resulted in retrieval of high in-fraction protein concentrations (Supplementary Fig. 1b) amenable to direct cryo-EM analysis. After SEC, we vitrified fractions 5 and 6 that include megadalton (MDa) assemblies (Fig. 1b, orange dotted line). In the acquired cryo-EM micrographs, large unperturbed complexes are visible, including fatty acid synthase (FAS)[44] (Fig. 1c) and polysomes (Fig. 1c) as well as prominent circular signatures (Fig. 1c). A frequent observation while analyzing the micrographs was the reoccurring proximity of the circular signature with filamentous structures and potential polysomes (Fig. 1d).

To first explore this signature, we collected 7011 movies at a pixel size of 1.57 Å and performed image analysis of the derived spherical signatures (Supplementary Fig. 2), resulting in an icosahedrally averaged cryo-EM map at 3.77 Å resolution (FSC = 0.143, Table 1) in which an initial backbone trace could be built. A search for structural relatives in the Protein Data Bank (PDB)[45] utilizing distance-based matrix alignments[46], identified the (only available) 3.5 Å *Saccharomyces cerevisiae* L-A virus crystal structure[34] as a potential match. Simultaneously, we used ModelAngelo[47], an automatic model-building tool, in its de novo mode, enabling us to construct an atomic model within the EM density. The patches generated in this process were then compared against the "non-redundant protein sequence (nr)" database, a task executed using the blastp suite[48]. The blast results confirmed our supervised identification of the *S. cerevisiae* L-A virus, showcasing impressive identification speed at this resolution regime (Supplementary Fig. 3).

Mass spectrometry (MS)-based protein identification of the fractionated cell extract verified the abundance of the L-A helper virus (Gag, Gag/Pol), and also the presence of the L-BC virus, albeit with lower abundance, after adding these endogenous protein sequences to the available yeast proteome (UP000002311)[22] (Fig. 1e, Supplementary Fig. 4). To annotate the proteomic content of the studied extract, an analysis of the measured biological triplicates of the two L-A virus-containing fractions was performed. Ribosomal proteins were identified with very high abundance after streamlining the classification of identified proteins using the Kyoto Encyclopedia of Genes and Genomes (KEGG) pathway database[49] (Fig. 1e, Supplementary Fig. 4). The capsid (gag) and the polymerase (pol) of the L-A virus were also identified within these fractions with high abundance (Fig. 1e, Supplementary Fig. 4). Compared to other studies, we succeeded in (a) mildly reducing sample complexity and rapidly retrieving L-A virus in yeast cell extracts directly amenable for cryo-EM; (b) identifying unambiguously the L-A virus using the high-resolution features of the calculated cryo-EM map by Cα-trace model building in a supervised and unsupervised manner; and (c) probing the environment and components of the L-A virus by quantitative mass spectrometry.

### The high-resolution cryo-EM structure of the L-A virus from a cell extract unveils structural and functional adaptations of the native state

To derive solid molecular insights into the L-A virus structure within the cell extract, we acquired an additional cryo-EM dataset at 0.93 Å. A high-quality de novo model was built in the derived 3.21 Å L-A virus cryo-EM map (FSC = 0.143, Table 1). The backbone was explicitly traceable (Fig. 2a); the α-helical pitch and β-strand separation were both discernable. Side chain densities were clearly visible, which allows us for the unambiguous placement of most side chains (Fig. 2b). The icosahedral protein shell has a triangulation number T = 1 organization, consisting of 60 structural units, with each unit comprising two chemically identical capsid proteins, formed by 120 protomers, with overall diameter (400 Å) and thickness (46 Å) comparable to its crystallographic counterpart (Supplementary Fig. 5a). A similar opening

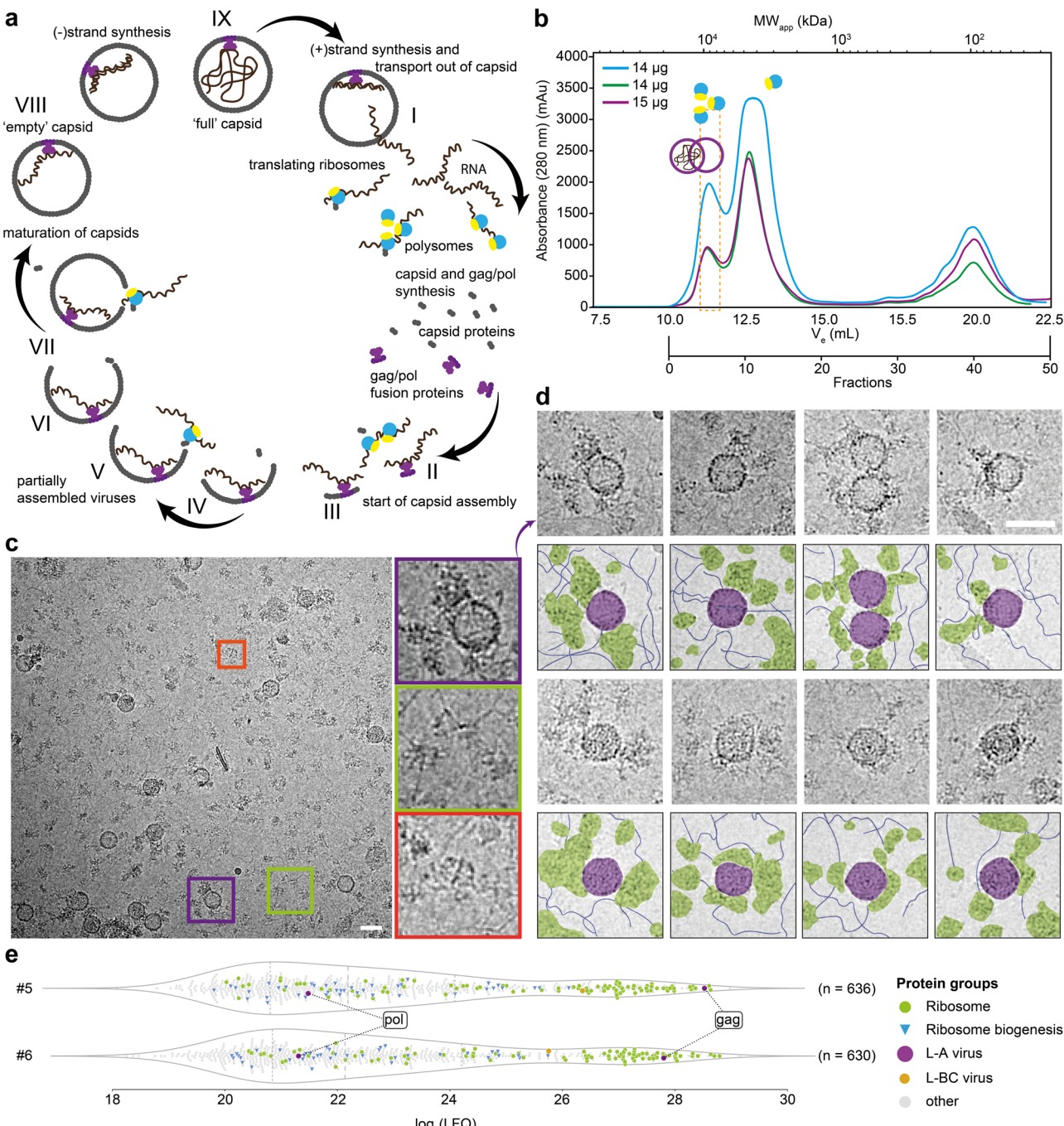

**Fig. 1 | Identification of the L-A virus in native cell extracts. a** Scheme of the yeast L-A virus life cycle (*states I-IX*). Replication starts with (+)strand synthesis and the transport out of the capsid (*state I*). Yeast ribosomes then translate the ssRNA and new viral particles (capsids) form (*states II–VII*). Last, the (-)strand is synthesized (*states VIII-IX*). Genome replication occurs in assembled viruses (*state IX*). **b** SEC profile of *S. cerevisiae* native cell extract. In this study, fractions 5 and 6, corresponding to ~10 MDa complexes, were further investigated (orange box). The purple circles show where full and empty L-A virus are expected based on relative retention. Accordingly, polysomes are also expected in the first peak and monosomes in the second peak. **c** Representative micrograph of fraction 5. A heterogeneous mixture of biomolecules can be seen, for example, the fatty acid synthase (red box), ribosomes (green box), and the L-A virus (purple box). **d** Representative crops of the L-A helper virus are shown in the vicinity of ribosomes, fibrillar structures, and other proteins. **e** LFQ-based MS quantification of fractions 5 and 6 to identify the L-A helper virus and the fraction contents. Scale bars represent 50 nm. n defines the number of identified proteins. Technical duplicates of biological triplicates were analyzed per fraction.

diameter (18 Å) at the icosahedral fivefold axes was observed that is hypothesized to serve as a gate for the exit of viral mRNA and entry of nucleotide triphosphates[34]. This highly conserved overall L-A virus architecture that is retained in both cell extracts and in the crystal points to high stiffness of the capsid and its openings, indicating critical structural rigidity.

Detailed analysis of the L-A virus identified previously undescribed stabilizing cation-π interactions in the capsid that mediate interactions within and also across L-A protomers (Fig. 2c). In the asymmetric unit, 26 side chains are involved in forming 11 distinct cation-π interaction networks, formed within monomeric and dimeric interfaces (examples are shown in Fig. 2c). Remarkably, residues involved in these interactions are

**Table 1 | Cryo-EM data collection, refinement, and validation statistics**

| | Acquisition 1 (viral abundance) (EMDB-15214) (EMPIAR-11069) | Acquisition 2 (Ribosome) (EMDB-15215) (EMPIAR-11069) | Acquisition 3 (L-A virus) (EMDB-15189) (PDB-8A5T) (EMPIAR-11069) | Acquisition 4 (L-A virus) (EMDB-17628) (PDB-8PE4) (EMPIAR-11581) |
|---|---|---|---|---|
| *Data collection and processing* | | | | |
| Magnification | ×45,000 | ×45,000 | ×92,000 | ×150,000 |
| Voltage (kV) | 200 | 200 | 200 | 200 |
| Electron exposure (e–/Å²) | 30 | 30 | 30 | 30 |
| Defocus range (µm) | −1.0 to -3.0 | −1.0 to −3.0 | −1.0 to −3.0 | −1.0 to −3.0 |
| Pixel size (Å) | 3.17 | 3.17 | 1.57 | 0.93 |
| Symmetry imposed | I | C1 | I | I |
| Initial particle images (no.) | 230,000 | 750.242 | 2.725.140 | 7.840.838 |
| Final particle images (no.) | 17,000 | 627,748 | 17,007 | 12,974 |
| Map resolution (Å) | | | | |
| FSC threshold | 6.4 (FSC = 0.143) | 6.62 (FSC = 0.143) | 3.77 (FSC = 0.143) | 3.21 (FSC = 0.143) |
| Map resolution range (Å) | | | | |
| *Refinement* | | | | |
| Initial model used (PDB code) | | | 1m1c | 1m1c |
| Map sharpening *B* factor (Å²) | | | 0 (not modified) | 0 (not modified) |
| Model composition | | | | |
| Non-hydrogen atoms | | | 10302 | 9963 |
| Protein residues | | | 1302 | 1255 |
| Ligands | | | 0 | 0 |
| *B* factors (Å²) | | | | |
| Protein | | | 76.63/187.82/113.84 | 13.52/149.90/80.27 |
| R.m.s. deviations | | | | |
| Bond lengths (Å) | | | 0.004 (0) | 0.005 (0) |
| Bond angles (°) | | | 0.985 (0) | 1.049 (1) |
| Validation | | | | |
| MolProbity score | | | 2.12 | 2.30 |
| Clashscore | | | 15.89 | 14.28 |
| Poor rotamers (%) | | | 0.45 | 1.78 |
| Ramachandran plot | | | | |
| Favored (%) | | | 93.68 | 92.9 |
| Allowed (%) | | | 6.32 | 6.86 |
| Disallowed (%) | | | 0 | 0.24 |

better resolved compared to side chains of the same type not involved in cation-π interactions (Fig. 2c, Supplementary Figs. 6–7, Supplementary Table 1). We deem the extended network of cation-π interactions observed to be highly specific (e.g., compared to non-specific hydrogen-bonding pattern) because a delocalized π-electron system must organize in the correct plane towards a cationic moiety. In the context of the L-A helper virus, cation-π interactions may contribute to both the folding of the monomeric building block and to the higher-order assembly of the virus itself.

The cation-π interaction represents a strong non-covalent bond[50,51]. To assess the prevalence of cation-π interactions across viruses, we developed a dedicated and highly specific software toolkit for their identification (see "Methods" for details). We analyzed the structural properties of icosahedral *Duplornaviricota* capsid structures (Supplementary Fig. 8). Notably, the *Totiviridae* family exhibited a pronounced prevalence of cation-π interactions. In contrast, such interactions were less common in the *Cystoviridae* family. This result points to a potential evolutionary diversification mechanism regulating aspects like protein folding and structural stability.

## Loop regions form a capsid-spanning flexible network and structure the mRNA decapping site

The crystallographically determined regions of the capsomere exhibit substantial conformational differences between the two monomers in the resolved asymmetric unit[34], i.e., residues N8–K12, G82, T96–I99, I111–T112, and G387–D396, which are also observed in the higher resolution cryo-EM model. Overall, the cryo-EM-derived asymmetric homodimer exhibits superior fits for both backbone ($CC_{cryo-EM}$ = 0.83 vs. $CC_{x-ray}$ = 0.76) and side-chain ($CC_{cryo-EM}$ = 0.82 vs. $CC_{x-ray}$ = 0.75) conformations (Supplementary Fig. 5b) which may be attributed to various factors, such as crystal packing.

Structural changes compared to the crystal structure are detected in both capsomere subunits (Supplementary Fig. 5b), specifically in loop regions (R94–V104; Y528–K536, G601–H607) of protomers A and B, respectively (Fig. 3a–d). Although all other loops could be clearly traced in the cryo-EM map those 3 regions had lower resolution, manifesting in diffuse electron optical densities. In detail, the protomer helix-turn-helix fold region R94–V104 forms the functional pores at the icosahedral five-fold

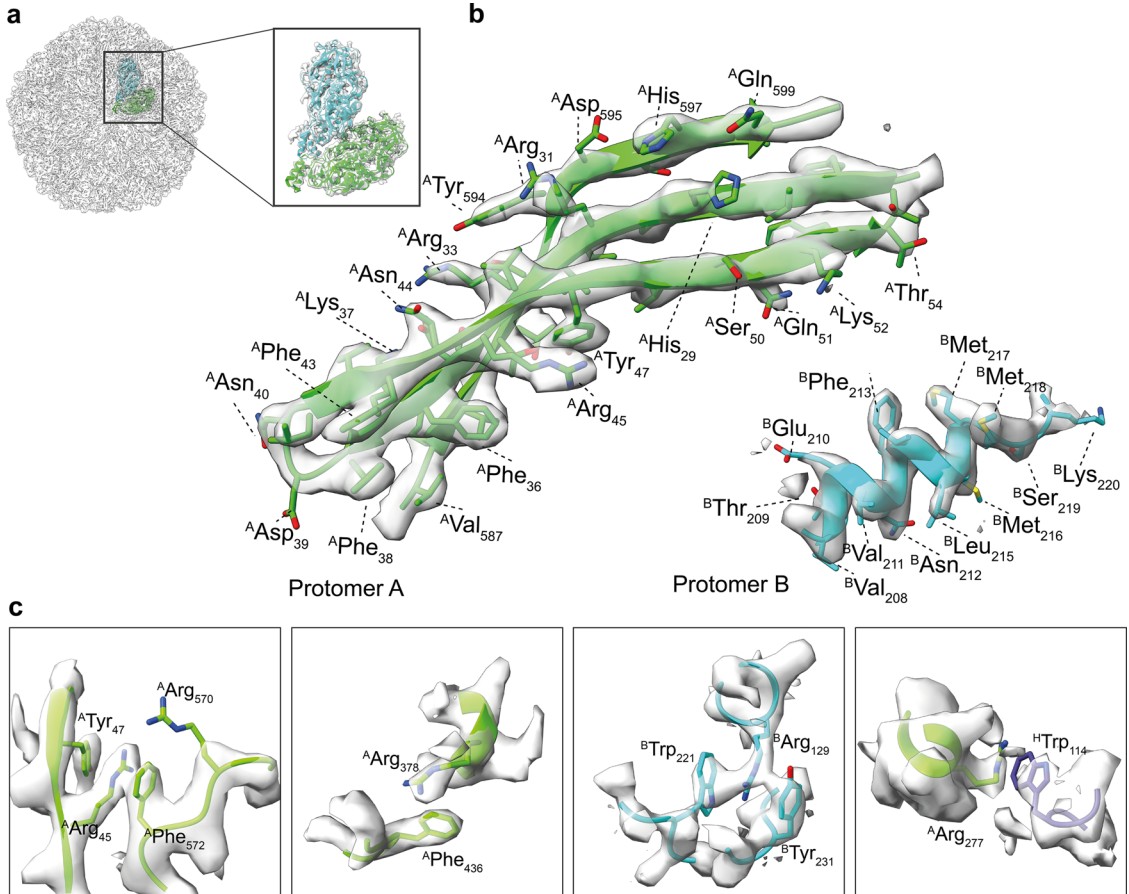

**Fig. 2 | Distinctive features stabilizing the L-A capsid. a** Derived cryo-EM map and model of the asymmetric subunit; the fitting is magnified in the inset. **b** Representative resolution features of the atomic model fitted in the 3.21 Å cryo-EM map. In green, a close-up of a β-sheet is shown with the separation of the single strands clearly visible in the density of the map; in blue, an α-helix is shown with helical pitch discernible; side chain density is often distinguishable. **c** Representative cation-π interactions are shown between arginine and tyrosine, tryptophan or phenylalanine within chain A (green), chain B (blue), or between chain A and an adjacent chain B (slate). The 3.21 Å cryo-EM density map is shown as transparent volume.

axes (Fig. 3b), supported by the exact same region in protomer B (Fig. 3b). Such a dual role (supporting and pore-forming) for this minimal fold may rationalize the observed flexibility in the cryo-EM structure. Next, loop Y528–L534 is near the active site His154 in each protomer (Fig. 3c), while loop G601–H607 is buried in distinct dimeric interfaces formed by the capsomeres (Fig. 3d, e). We compared the interfaces in which these loops are present to delineate their energetic contributions utilizing HADDOCK[52]. Their absence in the cryo-EM model as compared to their presence in the X-ray counterpart do not affect the overall energetics of the interfaces (Fig. 3). Rather, they seem to serve as regulatory regions. Flexible or intrinsically disordered regions, including loops, are known to contribute to the functional dynamics of proteins. They often act as molecular switches or hinges, enabling proteins to adopt different conformations and thereby play key roles in diverse biological processes. Given this, it is plausible to consider that the flexibility observed in these specific regions of the L-A virus could facilitate dynamic changes in the capsid structure under certain conditions, which might be pivotal for the virus's life cycle and function[53].

Extending energetic calculations to all proximal interfaces of the capsomere reveals interfaces of highly diverse energetic contributions where an interplay of van der Waals, electrostatics, and desolvation energies uniquely define each of the 9 surrounding interfaces as well (Supplementary Fig. 9a). Using these 9 unique interface energetic calculations, we propose a model of capsomere stability, which consists of 86 distinct steps of monomeric association to complete the capsid (Supplementary Fig. 9b; Supplementary movie 1). This analysis seems to suggest that capsid formation may occur via the sequential addition of monomeric subunits.

However, it's crucial to note that this is predicated on static model energetics and may not accurately depict the dynamic process of capsid assembly in vivo. Half-structures of the L-BC virus shows that capsid stability and assembly might be driven by clusters of monomers instead[54].

The L-A virus mRNA decapping activity is unique[34,36,38,39]. In both the x-ray and cryo-EM L-A virus structures, region Gln139-Ser182 (containing the active site His154 (Fig. 3f), contributes to the outer capsid surface, mediates cellular mRNA decapping, and transfers the 7-methyl-GMP (m[7]GMP) cap from the cellular mRNA 5'-end to the viral RNA 5'-end, countering a host exoribonuclease that targets uncapped RNAs. This active site was previously described to form a trench[55] that involves residues located in loop regions. Specifically, the flexible region spanning residues 528–534 (Fig. 3f) encircles the active site. This loop element undergoes dynamic changes as it was not Cα-traceable in the cryo-EM map; however, in the crystal structure it was resolved, pointing to a regulatory role in the binding site accessibility. Every monomer within the asymmetric unit forms an independent binding pocket at its active site; In the case of the X-ray structure, where this region is present, this flexible loop regulates the accessibility of this binding pocket.

## Detecting higher-order interactions of the L-A virus and minimal communities

Analysis of the lower pixel size (3.17 Å) cryo-EM acquisition to improve statistics (10,067 movies) shows the systematic proximity of electron-dense material to pleomorphic L-A viruses (Supplementary Fig. 10a). By 2D averaging picked L-A virus particles, 9711 single-particles with proximal

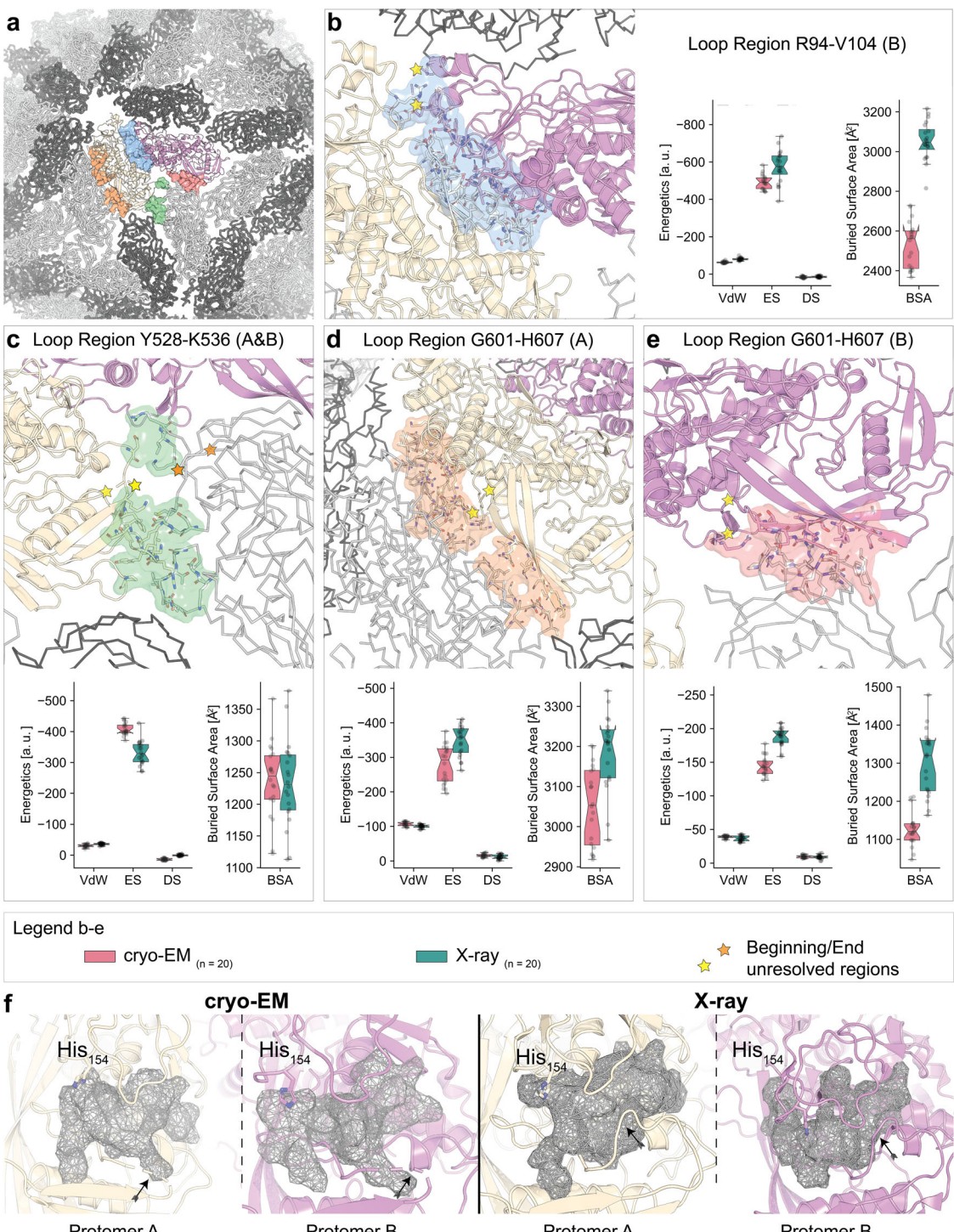

**Fig. 3 | Structural plasticity of the L-A virus capsid. a** An overview of the cryo-EM capsid structure. The asymmetric unit is highlighted in color and cartoon representation, while the remainder of the capsid is shown in ribbon representation. Dark gray indicates protomers A, light gray denotes protomers B. The unique interfaces involving the flexible regions in capsid-forming interactions are highlighted as transparent surfaces. **b–e** Interfaces from (**a**) are shown in more detail; residues around 5 Å of each interface are shown as sticks. The respective protomer is denoted in brackets in the panel title. The first and last resolved residues of the flexible linker regions are marked with yellow stars. Energetics were calculated with HADDOCK for the interfaces modeled from the 3.21 Å (red; missing the linker) cryo-EM map

and the 3.5 Å (green) X-ray counterpart. A total number of *n* = 20 models were refined for each condition and interface. **f** Comparison of the mRNA-binding active site pocket in the cryo-EM (left) and X-ray (right) structures. The flexible loop region Y528-K536 is highlighted by an arrow. In the X-ray structure, this loop confines the active site pocket while in the cryo-EM structure its corresponding density is of lower resolution, and therefore, flexible. The box minima represent the 25th percentile, the box maxima the 75th percentile, the Notch indicated the data's median, whiskers extend to the minimum and maximum value inside of a 1.5 interquartile range. All data points are overlaid as a scatter plot.

structural signatures can be derived, although the direct proximity must be highly flexible given the diffused densities (Supplementary Fig. 10b). In addition, a 3D reconstruction of ribosomal structure, resolved at 6.62 Å (masked, FSC = 0.143) from 627,748 particles (FSC = 0.143, Supplementary Fig. 11a, Table 1) corroborates the high abundance of ribosomes in the same fractions with the L-A virus. Overall, the 2D classification of picked single particles unambiguously shows that the cell extract fraction contains polysomes (Supplementary Fig. 11b) with a 3D conformation resembling published polysome conformations[56] and stalled disome states[57] (Supplementary Fig. 11b). From the ribosome 3D reconstruction (Supplementary Fig. 11a, 11c (left), Table 1) 3D variability analysis[58] was performed to further delineate ribosomal classes. Subsequent 3D reconstructions to resolve discrete heterogeneity across individual ribosomes reveal the presence of all major translation states with bound tRNAs at specific sites in the decoding center at resolutions from 10 to 13 Å (FSC = 0.143, Supplementary Figs. 11c and 12a–f), similar to the states derived from human ribosomes from multiple samples[59]. Overall, occupancies by tRNAs of the exit, peptidyl and aminoacyl sites are derived, as well as movements of the L1 stalk (Supplementary Fig. 11c). These results show that the L-A virus is present in a cell extract, in proximity to ribosomes captured in active translational states.

Identified L-A viruses within the cryo-EM micrographs, besides being in proximity to ribosomes and flexible macromolecules appeared pleomorphic (Supplementary Figs. 10a, c and 13). Their structural states resembled the states shown in Fig. 1a but statistical analysis of the native cell extract revealed that most particles identified are in their mature state (Supplementary Fig. 10c (I, VIII, IX), d, e) while other rare pleomorphic states were present (Supplementary Figs. 10c (II–VII) and 13).

All observed particles are in proximity to ribosomes and irregularly shaped macromolecules, possibly mRNA, and may be involved in, e.g., mRNA decaping considering the L-A virus capsid function. To check if these particles are broken during cell lysis, fractions were followed by western blotting against the L-A virus gag ("Methods"). Results show a high abundance of the virus in the high molecular weight SEC fractions, while, in the lower-molecular weight fractions, immunodetection of the viral capsid components did not yield a result (Supplementary Figs. 10f and 14) implying that the states observed could correspond to particles with minimal damage stemming from biochemical treatment. In addition, the distribution of pleomorphic states and their non-random distribution towards a higher abundance of empty and full viruses corroborates the visualization of relatively unperturbed particles (Supplementary Fig. 10d). The packaging efficiency of the virus, i.e., mature viruses encapsidating the genome, defined by the ratio between randomly counted full ($N = 400$) or empty ($N = 661$) mature capsids is ~40% (Supplementary Fig. 10e). At this resolution, we are unable to make any conclusions concerning the assembly mechanism[60], but our findings corroborate presence of immature viruses that are similar to those reported for the L-BC virus[54].

A common observation in the cryo-EM micrographs of L-A virus-containing yeast native cell extracts was formation of small groups of the L-A virus (Fig. 4a and Supplementary Fig. 15). In these groups, composed of $12 \pm 4$ viruses on average, the L-A virus is mostly in a mature form and in proximity to other proteins and RNA, but with the L-A virus communities separated from other single particles in the images (Fig. 4a). To investigate this observation, we imaged high-pressure frozen and vitrified 30 nm thick whole yeast cell sections by transmission electron microscopy. These sections show similar groupings (on average $16 \pm 10$ viruses, Fig. 4b). Distances between the closest viruses and their measured diameter are comparable statistically (Fig. 4c, d), corroborating their in-extract and in-cell correspondence. Such assemblies have been widely reported for exogenous viruses and are frequently associated with either membranes or other cytoskeletal features, for example being connected to the Golgi apparatus or exploiting actin filaments[3]. In our cell section images, the L-A virus communities show no obvious interactions with the cytoskeleton or membranes but colocalize with the translation machinery, i.e., mostly ribosomes and irregular, electron-dense macromolecules that could represent mRNA

(single-particle micrographs). These viral communities appear to be nano compartments that are relatively isolated from other cellular material.

## Discussion

Endogenous viruses have often been overshadowed by their exogenous counterparts which are known for causing the rapid spread of diseases, such as Ebola, SARS-CoV1/2, and HIV. However, endogenous viruses, including human endogenous retroviruses (HERVs) and human mammary tumor virus (HMTV), hold importance for understanding virus-host interactions and their potential impacts on host health[61]. These endogenous viruses are less understood due to their unique mechanisms of pathogenicity, which can differ from those of exogenous viruses. They can be inherited, making their potential impact on offspring's health a crucial area of study. In fact, accumulating evidence suggests that HERV aberrancies might be associated with complex disorders like schizophrenia, characterized by the interplay between genetic and environmental factors[62].

Using the L-A virus as a model, we show that this does not happen at random cytoplasmic regions, but in communities that may involve different steps in capsid maturation, associated with the translation apparatus.

Our work also reveals critical insights into virus structure in the endogenous context: The reconstruction of the L-A viral capsid from native cell extract using cryo-EM attains a resolution higher than the previously published, highly purified structure using x-ray crystallography[34] and stems from a sample which is captured in a highly heterogeneous environment, interacting with other electron-dense cellular material (e.g., compare Fig. 1c to Fig. 1 from Castón et al.[33]).

Unsurprisingly, the core regions of the capsid do not show any major differences, but the solvent-accessible flexible loop regions show clear deviations, which may represent functional differences in the native environment. Interestingly, multiple cation-π interactions are identified within the capsid subunits that are not only major contributors to the stability of the capsid but also may assist in folding and higher-order assembly. This is because a cation-π interaction needs the exact placement of both partners, i.e., the delocalized π-electron system and the cationic moiety. Once formed, this interaction is rigid and has clearly visible well-resolved side chains. We observed these interactions to be widespread in *Totiviridae* and do not appear as frequent in other double-stranded RNA viruses, highlighting a possible evolutionary divergence mechanism at the molecular level.

As previously described, the L-A capsid harbors a decapping and 'cap-snatching' activity[34,38,39], located at the capsid surface. By this, the virus can (a) cap its own mRNA for increasing the half-life, and (b) create mRNA decoys for degradation. An adjacent flexible loop forms the active site pocket and regulates its accessibility. This structural feature may control the interaction of endogenous mRNA and thus likely plays a role in the cap-snatching activity of the capsid.

Overall, by analyzing the L-A virus in its near-native environment, we were able to identify another layer of the viral architecture. By forming communities, the newly transcribed viral mRNA is immediately translated by the associated ribosomes, forming clearly visible polysomal structures at the site of capsid formation. By decorating the mRNA with a tight layer of ribosomes, mRNA degradation cannot take place. Putting the 'cap-snatching' mechanism in this context, capping the viral mRNA might not primarily serve as protection, but as an enhancer for translation of the viral proteins.

## Methods

### Cell cultivation and lysate fractionation

The *Saccharomyces cerevisiae* strain from ATCC (American Type Culture Collection PO Box 1549 Manassas, VA 20108, USA; ATCC® 24657™) was cultivated in YPDG medium at 30 °C for 5 h, to a $OD_{595}$ of 2.5 (early exponential phase), then harvested at $3000 \times g$ for 5 min at 4 °C. The pellet was washed by resuspending in distilled water and recentrifuged with the same parameters (resulting pellet ~7 g). The resulting pellet was then lysed via bead beating using glass beads and lysis buffer (100 mM HEPES pH 7.4, 95 mM NaCl, 5 mM KCl, 1 mM MgCl₂, 5% Glycerol, 0.5 mM EDTA, 1 mM

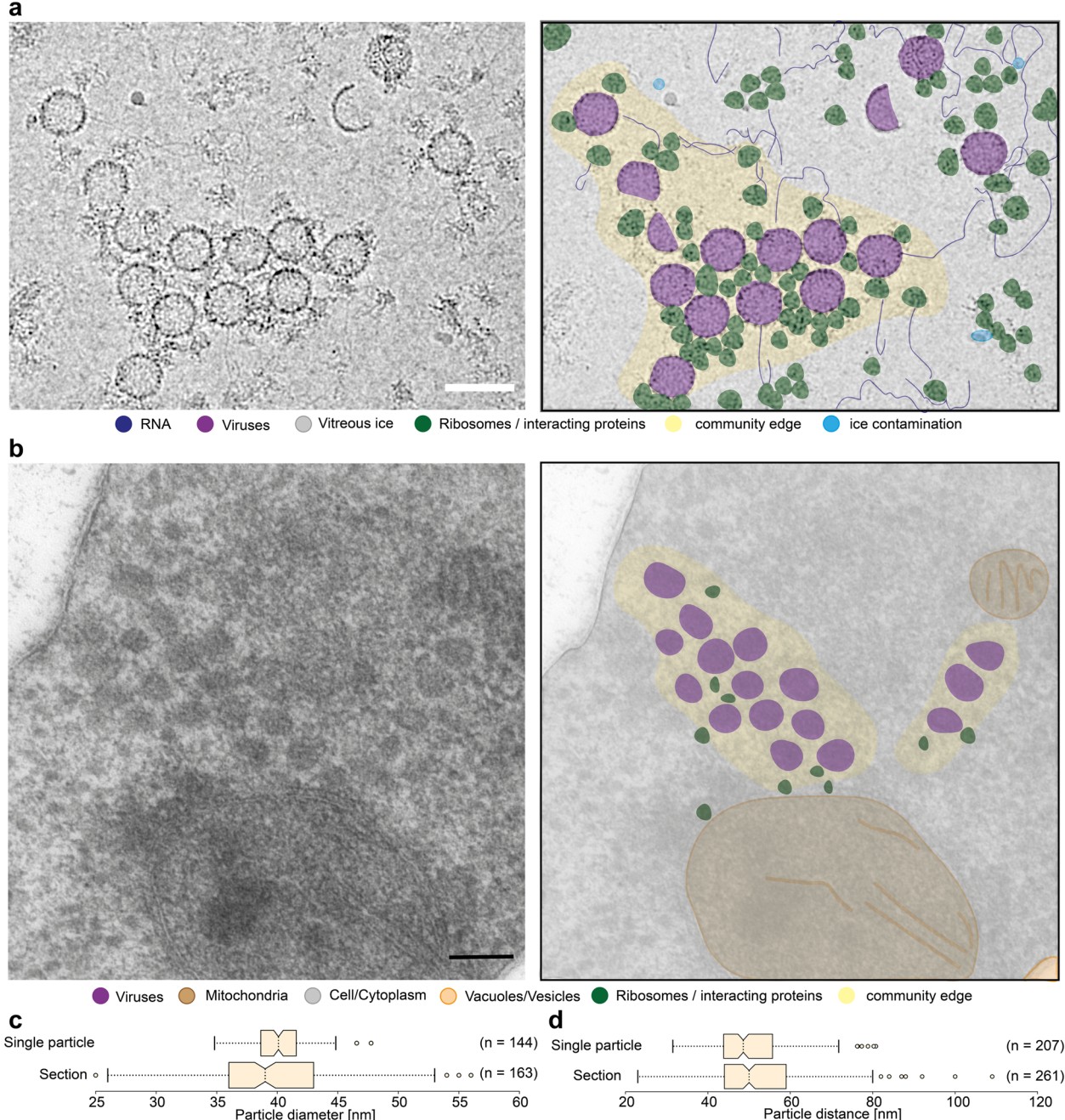

**Fig. 4 | L-A virus communities in native cell extracts and in yeast cells. a** Proposed viral communities are shown in a crop of a micrograph. Molecules are colored, such as RNA in dark blue, ribosomes in dark green, ice contamination in light blue and L-A virus in purple. The space around the group of viruses and surrounding proteins is colored yellow. **b** Cryo substituted, in resin embedded *Saccharomyces cerevisiae*, cut into sections of 30 nm in width and imaged on the EM 900. Shown is a crop through yeast on the left and an impression on the right. Color coding stays the same as in (**a**). The measured particle diameter (**c**), as well as the distances between L-A viruses (**d**), are represented by box plots with the standard deviation shown in the whiskers. All scale bars represent 50 nm. **c, d** The box minima represent the 25th percentile, the box maxima the 75th percentile, the Notch indicated the data's median, whiskers extend to the minimum and maximum value inside of a 1.5 interquartile range. Outliers are indicated by scatters and sample size is indicated. The *n* values indicate the number of individual measured particles (**c**) and number of unique particle pair distances (**d**).

DTT, 10 µg/ml DNAse, 2.5 mM Pefabloc, 40 µM E-64, 130 µM Bestatin, 0.5 µM Aprotinin, 1 µM Leupeptin, 60 µM Pepstatin A) in 3 rounds for 20 s of 6.5 mps in a fast prep. The lysate was centrifuged for 5 min at $4000 \times g$ at 4 °C (Beckmann Heraeus multifuge 16R Swinging bucket rotor). The supernatant was again centrifuged at $100,000 \times g$ for 45 min at 4 °C (Beckman Optima™ MAX-XP Rotor MLS-50). After filtering the supernatant through a 0.22 µm syringe filter, the filtered lysate was concentrated using an Amicon with a cut-off of 100 kDa. Protein concentration was measured using Bradford reagent.

The lysate (concentration of 14 mg/ml) was loaded on a size exclusion column type Biosep 5 µm SEC-s4000 500 Å LC- Column 7.8 × 600. The system used was an ÄKTA Pure 25 M FPLC; the fractionation was performed at a 0.15 ml/min flow rate, with a 200 mM $NH_4CH_3COO$-buffer pH 7.4 (filtered and sonicated), and fractions of a total volume of 250 µl were collected (Fig. 1b). The eluted fractions concentration was determined using Bradford reagent and directly used for data acquisition or flash-frozen and stored at −80 °C until further use.

## Sample preparation for cryo-EM and data collection

To prepare samples for cryo-electron microscopy, holey carbon-coated copper grids (type R2/1 200 mesh from Quantifoil) are glow discharged using a PELCO easiGlow (15 mA, grid negative at 0.4 mbar, and 25 s glowing time). Directly after glow discharging the grids, 3.5 µl of the sample at a concentration of ~0.3 mg/ml is applied to each grid and plunge frozen using the Vitrobot® Mark IV System (Thermo Fisher Scientific) and standard Vitrobot Filter Paper (Grade 595 ash-free Ø55/20 mm). Vitrification conditions were as follows: humidity 100%, temperature 4 °C, blotting time 6 s, blot force 0, sample volume 3.5 µl, blot offset 2 mm. After clipping, the vitrified grids were loaded onto a Thermo Fisher Scientific Glacios 200 kV Cryo-transmission electron microscope. The Falcon 3EC and the Falcon 4i direct electron detectors were used for acquisition. The images were acquired in linear mode with a total dose of 30 e⁻/Å² (the acquisition parameters are shown in Table 1).

## Image processing, analysis, and model refinement

Three datasets were acquired for this work. Two datasets of fractions 5 and 6 at a pixel size of 3.17 Å (Acquisition #1 and #2), a smaller one with 2 replicates of fraction 5 with 1.57 Å (Acquisition #3), and a high-resolution dataset of fraction 5 with a pixel size of 0.938 Å (Acquisition #4). Acquisition #1 to #3 were acquired using a Falcon III direct electron detector in mrc-file format, while Acquisition #4 was recorded using a Falcon IV direct electron detector and eer-file format (see Table 1). Acquisition #1 was first analyzed with Scipion3[63–66] and the refined map was then transferred to cryoSPARC[67] for further analysis. The two other acquisitions were processed using cryoSPARC only. All movies were firstly motion-corrected[68] and then CTF[69] estimation was performed before particle picking. For the L-A viral particle, a small subset of micrographs were manually picked to generate suitable templates. These templates were used for template picking in both acquisitions, using the Falcon 3 and Falcon 4 camera. For the ribosomal dataset, a blob picker with a diameter range of 550 to 650 Å was used. All datasets were subjected to sequential rounds of 2D and 3D classification. After removing low-resolution data, final datasets of 17,000 particles (L-A virus; Falcon 3; #1), 627,748 particles (Ribosomes; #2), 17,007 particles (L-A virus; Falcon 3; #3), and 12,974 particles (L-A virus; Falcon 4; #4) were used for final 3D reconstruction. The resulting maps were refined, by applying icosahedral symmetry and symmetry expansion in cryoSPARC[67]. The FSC was calculated and the local resolution was derived according to the gold standard FSC[63]. The icosahedral averaged map from acquisition #1 containing two independent protomers was of sufficient quality to build a Cα trace using Coot[70]. A subsequent DALI search[71] identified the X-ray structure of the L-A virus (PDB ID: 1m1c) as best hit. Thereafter, the model was rigid-body fitted into the density using ChimeraX[72] and subjected to iterative refinement cycles using Coot[70] and PHENIX[73] yield the final coordinates. The PHENIX tool "Comprehensive validation (cryo-EM)" was used with default parameters[74] to calculate map cross-correlation coefficients. RMSD between structural models were calculated per residue by custom python code.

## Mass spectrometric analysis

In-solution digestion was performed for sample preparation for mass spectrometric (MS) analysis. 10 µg of fraction 5 or 6 (technical duplicates of biological triplicates) were precipitated and adjusted to 100 µl with ice-cold acetone and incubated for 60 min at −20 °C. Precipitated proteins were centrifuged (10 min, 20,000 × g) and air-dried. Protein pellets were resuspended in 25 µl of 8 M urea in 0.4 M ammonium bicarbonate, reduced with 5 µl of 45 mM DTT (30 min at 50 °C), and alkylated with 5 µl of 100 mM 2-chloroacetamide (30 min at 37 °C). The volume of each sample was adjusted to 200 µl by water. The samples were digested using trypsin (Promega Sequencing Grade Modified Trypsin) using a 1:50 (w/w) enzyme:protein ratio for 16 hours at 37 °C. The reactions were stopped by adding 10 µl of 10% (v/v) TFA. 20 µl of digestion mixtures were analyzed by LC/MS/MS using a U3000 nano-HPLC system coupled to a Q-Exactive Plus mass spectrometer (Thermo Fisher Scientific). Peptides were separated on reversed phase C18 columns (trapping column: Acclaim PepMap 100,

300 µm × 5 mm, 5 µm, 100 Å, Thermo Fisher Scientific; separation column: µPAC 50 cm C18, Pharmafluidics). After desalting the samples on the trapping column, peptides were eluted and separated using a linear gradient ranging from 3% to 35% B (solvent A: 0.1% (v/v) formic acid in water, solvent B: 0.08% (v/v) formic acid in acetonitrile) with a constant flow rate of 300 nl/min over 180 min. Data were acquired in data-dependent MS/MS mode with higher-energy collision-induced dissociation (HCD), and the normalized collision energy was set to 28%. Each high-resolution full scan (m/z 375 to 1799, R = 140,000 at m/z 200) in the orbitrap was followed by high-resolution fragment ion scans (R = 17,500) of the 10 most intense signals in the full-scan mass spectrum (isolation window 2 Th); the target value of the automated gain control was set to 3,000,000 (MS) and 200,000 (MS/MS), maximum accumulation times were set to 50 ms (MS) and 120 ms (MS/MS). Precursor ions with charge states <2+ and >6+ or were excluded from fragmentation. Dynamic exclusion was enabled (duration 60 s, window 3 ppm).

## Identification of most abundant proteins from MS/MS data

MS-raw files were analyzed with MaxQuant (version 1.6), with activated label-free quantification (LFQ), iBAQ and "Match between runs" option, and the annotated yeast proteome (UP000002311). For identification of viral proteins, the sequences of the following L-A virus proteins were added to the database: gag (UniProt-id: P32503) and pol (UniProt-id: Q87022; residue-range: 647–1505), the L-BC gag-pol (UniProt-id: P23172), the ScV-M1 preprotoxin (UniProt-id: P01546), the ScV-M2 preprotoxin (UniProt-id: Q87020), and the ScV-M28-like killer preprotoxin (UniProt-id: Q7LZU3). MaxQuant derived results files were analyzed and plotted by an in-house python script.

## Energetic calculations

The guru- and multi-body-interface of the HADDOCK webserver 2.2[52,75] was used to calculate protein–protein interaction energetics. All clustered models were used to calculate the energetic distribution. For energetic calculation, the model refined in the 3.77 Å was used. Calculations were done using an in-house python script. For calculations of capsid stability, the capsid structure was used as a template and, starting from a monomeric subunit, the energetics for each possible monomeric interaction partner was calculated, using the 9 unique interfaces of the capsomere (see Supplementary Fig. 9), and the formula:

$$E_{total} = \sum_{i=1}^{n}(E_{vdw}(i) + E_{es}(i) + E_{ds}(i))$$

Best ranked "docked" monomer were selected. This was repeated, till the entire capsid was built. Calculations were done by a PyMOL python script.

## Western blot analysis

Gels were freshly casted in-house prior to the experiment using a separating gel: 10% (w/v) acrylamide (37.5:1), 0.1% (w/v) SDS, 0.04% (w/v) APS, 0.002% (w/v) TEMED in 370 mM Tris-HCl solution pH 8.8 and stacking gel: 5% (w/v) acrylamide (37.5:1), 0.1% (w/v) sodium dodecyl sulfate (SDS), 0.04% (w/v) APS, 0.002% (v/v) TEMED in 125 mM Tris-HCl-solution pH 6.8. The gels cast were 1 mm thick. The samples were mixed with a 4x loading dye (250 mM Tris-HCl (pH 6.8), 8% w/v SDS, 0.2% w/v bromophenol blue, 40% v/v glycerol, 20% v/v β-mercaptoethanol) and incubated for 5 min at 95 °C shaking. Roughly 3 µg sample of the fractions and 2.5 µg as well as 5 µg of the positive control, was loaded and electrophorized with a standard of 5 µl of Precision Plus Protein™ All Blue Prestained Protein Standards (BioRad #1610373). For electrophoresis, a 1x electrophoresis buffer freshly prepared using a 10x stock solution (30.3 g Tris-base, 144 g Glycine in 1 L of deionized water) was used and the gels run at an electrical field of 100 V for ~2 h. The gels were then transferred onto a nitrocellulose membrane using the Trans-Blot® Turbo™ Transfer System of BioRad. A preset protocol of 25 V (1 A) applied field, for 30 min was used. After blotting

the membranes were blocked using 5% (w/v) skimmed milk powder in TBST for 1 h at 4 °C. The primary antibody (rabbit anti-ScV-L-A peptide serum by GenScript) was diluted at 2:25,000 in 2% (w/v) skimmed milk powder in TBST and incubated for 16 h at 4 °C. After incubation, the membrane was washed 3 times for 10 min with 2% (w/v) skimmed milk powder in TBST. The secondary antibody (goat anti-rabbit IgG H&L (HRP) ab205718 by Abcam) was prepared the same way as the primary but at a final concentration of 1:25,000. The membranes were incubated for 1 h at RT. After another 3 washing steps, the membrane could be imaged using the ChemiDoc MP Imaging system and a freshly prepared ECL fluorescent mixture. Antibodies were custom-made by GenScript (New Jersey, USA).

## Automated model building
For automated model building, the asymmetric unit of the capsid's density map was isolated using PHENIX 'Map Box'[73] tool by default. This asymmetric map served as the foundation to run ModelAngelo[47] with the 'build_no_seq' flag. The resulting model was analyzed using PyMOL, and sequences of patches longer than 20 residues were isolated. These sequences were then blasted against the non-redundant sequence dataset (nr) using the blastp webserver[48]. All results underwent manual curation.

## Cation-π interaction identification
To identify Cation-π interactions in the atomic structures, the self-developed CatPiToolkit was utilized as the default setting. This toolkit measures the distance between the cationic side chain moiety and aromatic side chains. A potential cation-π interaction is recorded if the cation aligns perpendicularly to the aromatic ring, and the distance is within a defined threshold (default set at 5 Å).

All identified interactions were manually curated and validated. Resolvability threshold for all residues was established using a custom Python script executed in ChimeraX[72]. The outcomes were subsequently analyzed. Statistical significance of the findings was computed using a two-sample t-test, facilitated by the Python SciPy library[76].

## Pocket identification
The analysis of the active site pocket was conducted using PyVOL[77] in its default settings.

## High-pressure freezing
Yeast cells (harvested as described in the section Cell cultivation and lysate fractionation) were rapidly frozen with a high-pressure freeze fixation apparatus (HPM 010; BALTEC, Balzers, Liechtenstein). The material was cryo substituted with 0.25% glutaraldehyde (Sigma) and 0.1% uranyl acetate (Chemapol, Prague, Czech Republic) in acetone for 2 days at −80 °C using cryo substitution equipment (FSU; BAL-TEC) and embedded in HM20 (Polysciences Europe) at −20 °C. After polymerization, samples were cut with an ultramicrotome (Ultracut S, Leica, Wetzlar, Germany). The ultra-thin sections (30 nm) were transferred to formvar-coated copper grids and poststained with uranyl acetate and lead citrate in an EM-Stain apparatus (Leica) and subsequently observed with a Zeiss EM 900 transmission electron microscope (Carl Zeiss Microscopy GmbH, Jena, Germany) operating at 80 kV.

## Statistics and reproducibility
Gold standard FSC was used to derive cryo-EM reconstructions. This randomization involves separation of particles in two sets that are reconstructed independently.

## Reporting summary
Further information on research design is available in the Nature Portfolio Reporting Summary linked to this article.

## Data availability
The 3D maps of the L-A virus at 3.21 Å (PDB 8PE4; EMD-17628) and 3.77 Å (PDB 8A5T; EMD-15189), and the in-extract ribosome (EMD-15215) are available at PDB and EMDB databases, respectively. The proteomics dataset can be accessed through PRIDE (Identifier: PXD034431). Source data are provided with this paper and all other data for support of this study are available upon request. Primary cryo-EM movies are deposited in EMPIAR (EMPIAR-11069 and EMPIAR-11581). The numerical source data for graphs and charts in the manuscript can be found in the Supplementary Data file. Any remaining information can be obtained from the corresponding author upon reasonable request.

## Code availability
The toolkit to unambiguously identify potential cation-π interactions and its documentation can be accessed on GitHub at https://github.com/kastritislab/CationPiToolkit[78].

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

## Acknowledgements
The authors thank all members of the Kastritis laboratory for valuable insights, in particular Kevin Janson for growing yeast and Ioannis Skalidis for initially setting up data analysis of cryo-EM acquired fractions. The authors also thank Dr. Marta Fratini, and Prof. Dr. Ingo Heilmann (MLU, Halle, DE) for confocal microscopy of yeast protoplasts, as well as Prof. Dr. Stephan Feller and Dr. Lolita Piersimoni for valuable discussions. A special thanks for Dr. Pranav Shah and Prof. David I. Stuart for the valuable feedback. This work was supported by the Federal Ministry for Education and Research (BMBF, ZIK program) (Grant nos. 03Z22HI2 and 03Z22HN23 to P.L.K. and 03COV04 to P.L.K. and M.T.S.), Horizon Europe ERA Chair 'hot4cryo' project number 101086665 to P.L.K., the European Regional Development Funds for Saxony-Anhalt (grant no. EFRE: ZS/2016/04/78115 to P.L.K. and M.T.S.), funding by Deutsche Forschungsgemeinschaft (DFG) (project number 391498659, RTG 2467), and the Martin-Luther University of Halle-Wittenberg.

## Author contributions
L.S. performed sample preparation, cryo-EM grid preparation and screening, and F.L.K. supervised the laboratory work. F.H. collected the data. EM-Data analysis and map calculations were performed by L.S., D.A.S., C.T. and P.L.K. M.T.S. build the initial model and identified the L-A virus via the DALI search. Structural modeling and structure validation were performed by C.T. M.S. analysis was performed by L.S. and C.T. C.I. measured MS data using A.S. instruments. G.H. performed cryo substitution experiments. A.M. contributed to the conceptualization of the project. C.T. and P.L.K. wrote the paper, with contributions from L.S. and all authors; L.S., C.T. and P.L.K. made the figures. P.L.K. conceived, funded, and supervised the project.

## Funding

## Competing interests
The authors declare no competing interests.
