## [Peer Review File · Communications Biology]

Reviewers' comments:

Reviewer #1 (Remarks to the Author):

The manuscript describes cell-extracted particle structures of yeast Sc-L-A virus. Only 3.4-Å X-ray crystal structure of the L-A virus was available. The significance of the study was to determine an atomic model of near native structure of cellular L-A virus particles. The most of so-far published structural studies of viruses focused on resolving a matured virus structure in extracellular phase. First, it is shown that mass spectrometric and morphological characterizations of the L-A virus particles and associating polysomes in yeast lysate fractions (Figs. 1 and 6, Suppl. Figs. 1, 10-15). Second, it is resolved a near-native structure of the matured yeast Sc-L-A virus particles at the resolution of 3.7-Å and discussed possible interactions and methyl-transferase active site in the capsid (Figs. 2-4, Suppl. Figs. 2-6). Third, inner capsid structure is described at a low resolution (Fig. 5, Suppl. Figs. 7-9). Based on these findings, limited but new structural properties of the cellular L-A virus capsid seem to be revealed. However, I am skeptical to the significance and the purpose of studying “endogenous” structure of the L-A virus, and doubt an interpretation of the results, especially second and third findings.

Main comments

1. Introduction P3 L54-75, P4 L108-L117, Discussion P11 L391-404

I doubt the Sc-L-A virus is an adequate target for fulfilling the significance since the Sc-L-A virus only infects intracellularly (endogenous), not extracellularly (exogenous), although I agree on the significance of resolving structure of the endogenous virus particles. The significance of the study is overinterpreted in “Introduction” and “Discussion” sections. Strictly speaking, a major contribution of the manuscript is to determine the first cryo-EM capsid structure and the atomic model of the matured L-A virus. The structural differences between former X-ray structure and the cryo-EM structure in this manuscript should not be due to the ones between endogenous and exogenous particles. The authors describe COVID-19 and other human pathogens as examples in Introduction and Discussion section, however they have both intracellular and extracellular phases and are enveloped ssRNA viruses, and thus they do not akin at all to the dsRNA non-enveloped yeast Sc-L-A virus with only intracellular phase. The same idea and methods cannot be applied for these intracellular/extracellular human viruses. I strongly recommend focusing on intracellular matured dsRNA L-A virus structure and reducing largely the description in Introduction and Discussion sections.

2. Results P6 L180 – P7 L228, Fig. 2, Suppl. Fig. 3-4, Discussion P11 L413 – L418

I am not convinced the description with regards to side chain interactions and conformations at the resolution of 3.7-Å. The achieved resolution of the cryo-EM model (3.7-Å) is worse than available crystal structure (3.4-Å). In this resolution, it is very difficult to interpret the side chain interactions such as cation- π stacking and side chain conformational variations. The atomic model is largely biased by the achieved resolution of the cryo-EM map and the side chains can be overfitting to the map (This can also be a reason of cc value is unexpectedly high for 3.7-Å cryo-EM map in suppl. fig. 4.). As shown in Fig. 2, the side chains look like a bump and it is difficult to determine the accurate orientations. However, it

might be possible to compare the experimentally obtained cryo-EM map (3.7-Å) with a calculated density map from the crystallographic atomic model at a 3.7-Å. It should be justified the descriptions by considering the achieved resolution.

3. Results P7 L230 – P8 L251, Discussion P12 L420 – L432

It is not easy for me to see the differences of the flexible loops between X-ray and cryo-EM structures in Fig. 4 and seems to be quite limited differences. It looks both structures show an opened conformation. For example, in main text (P7 L245), the authors mention, “their distance is 3-Å closer in the cryo-EM structure”, however the achieved resolution is 3.7-Å, which cannot interpret the structural difference of the 3 Å. It is too speculative that these additional flexible loops contribute to open and closed states of capsid methyl transferase active site. Again, X-ray and cryo-EM structures should not reflect the differences of exogenous and endogenous structures. I do not think the cryo-EM structure shows more functional conformation except crystal packing.

4. Results P8 L265 – P9 L316, Fig. 5

The analysis of this part has not been performed well technically and is not reliable. I strongly suggest removing this part fully from the manuscript. It is not described the details at all how to reconstruct the interior structure in Methods section (C1 reconstruction). The 2D-class averaged images after subtracting capsid intensity contribution are very strange (Fig. 5A). The intensity of the capsid is still largely remained after the subtraction. In Fig. 5B and 5C, why is only a portion of the pol/genome density revealed? The viral genome density should be distributed more evenly in the capsid. The obtained map might be pol or viral genome or something else at the low resolution. The AlphaFold2 prediction does not help for determining the position of the capsid and the pol since there is no reliability of positioning the capsid and pol as clearly shown in results in suppl. fig.8b (The linker region is poorly predicted in green region, left figure. No reliability of positioning capsid and pol shown in right figure).

Minor comments

Introduction

P3 L54-55 How does the structural analysis of viruses contribute to epidemiology and ecology? It may be, but it is a confusing description.

P3 L55-56 Virus diagnostics (virus detections in cells) were traditionally performed serologically and genetically. Or, the authors means morphological characterizations of viruses?

P3 L72-75 I do not understand the intended meaning. Please clarify.

P4 L97 Double-stranded ds(RNA)

P4 L109 “after successful biochemical enrichment”. This description is clearly too much. What author did was to lyse infected yeast cells and to purify virus fractions using conventional size exclusion column purification, not biochemical enrichment. It may mean MASS analysis, but then please say so.

Results

P5 L121 “cryo-EM accessible” Cryo-EM can be accessible to varieties of specimens including cellular samples. It is a confusing description.

P5 L134, Fig. 1c How is the complex confirmed to be a FAS in a raw cryo-EM image?

P5 L159-160 I do not think that enriching the L-A virus is unique to this study compared with other studies. The beads beating cell lysis and fractionations using SEC is a regular method for many other studies.

P6 L173 I understand that it has often describe to be T=2 in some icosahedral dsRNA viruses since they have two identical subunits in an asymmetric unit. However, T=1 should be a correct description. T=2 icosahedron does not mathematically existed according to the definition of T number ($h^2 + hk + k^2$).

P7 L226-228, Suppl. Fig. 5b I may not fully understand the analysis, however if A-B dimer is formed initially, all capsid protein form the A-B dimer for assembling? If so, then monomer subunits do not exist a lot? I am not really believed that monomer units are one by one gathered sequentially to make a capsid.

P8 L252-264, Suppl. Fig. 6 There is no description of DALI score in the main text and figure legend. Are they sufficiently reliable to say distant structural homologs? According to a DALI developer’s instruction, a Z-score between 2 and 8 is a grey area. It is not convincing to say “as well as structural homologs with potential evolutionary implications” in L263-264. It is still very speculative and very challenging to find its evolutionary relation to other methyl transferases.

P10 L336-344 I am not an expert of ribosome structures, however I feel it is needed to show structural differences of the 3D reconstructions of the active ribosomes. In the intermediate resolution (8-10 Å) in suppl. fig.11c, is it possible to distinguish their different active states including tRNA occupancy?

P11 L376 – L379, Fig. 6b How was a segmentation of the image done for viruses and ribosomes/interacting proteins in the entire cell image (Fig. 6b)?

Methods

P13 L477 – 479 Any energy filter was used for improving obtained images of thick virus samples?

P13 L483 – P14 L510 It is straightforward and sounds better to put accurate particle numbers in this section, e.g. “approximately 230000 particles” should be the exact number.

Reviewer #2 (Remarks to the Author):

In this manuscript Schmidt et al., used single-particle cryo-EM to visualize the yeast L-A endogenous virus in near native environment. Major finding of this study is the ability to use cryo-EM to visualize the virus in its native state in a cell and show interaction with cellular machinery (ribosomes). It is surely a technological advancement and will be of interest to wider field.

Authors were able to determine the structure of L-A capsid to a modest resolution of 3.77Å and build a c-alpha model. Even though the crystal structure of capsid has been previously solved, authors found some novel interactions and dynamic flexibility in their cryo-EM structure.

Even though the technological advancement is obvious, it remains to be seen if this approach can identify a novel protein correctly. Most of the biological findings in the study are not novel and have been reported earlier that viral replication involves concentration of cellular machinery named "factories".

Authors should speculate the feasibility of this approach to determine the structure of a unknown protein interacting with viral components if the X-ray structure is not available for unknown protein.

Additional comments-

Page 5- line 130 and Page 13, line 443. Please elaborate the washing of cell pellet with water. Was it resuspended in water and spun or just rinsed in water?

Page 5, line 157- Do authors mean there were 1.8 +/- 0.49 polymerase present per capsid? That would be significant more than 60:1 ratio of gag to gag-pol ratio. Ref 31 had suggested 1.9% of ribosomes producing gag-pol after -1 frameshifting.

What is the biological significance of cation-pi interactions identified in this study (Fig 2). It needs to be confirmed by site-directed mutagenesis.

Page 9- line 293 describes Fig 5d but I could not find Fig 5D in figure.

How were the ribosome particles picked (manual, blob, template or combination thereof ?) during the cryo-EM processing (Supp Fig 11). Its description was missing from methods. It will help wider cryo-EM community to pick particles of similar dimension containing different viral (capsid here) or cellular macromolecule targets.

Page 10- line 348-349. How do authors consider the fact that partially assembled viral particles (stage II-VI in Fig 1a) will not be as electron dense as the whole particle and that will make picking them from micrographs unreliable. Unless the viral replication is synchronized by external factors there should be broad heterogeneity in viral particles and majority may not be in mature state.

Page 14- line 504, Coot is not a manual refinement program. Appropriate word may be "interactive refinement".

Reviewer #3 (Remarks to the Author):

In this work, an icosahedral reconstruction and an asymmetrical reconstruction of the L-A virus retrieved from cell extracts at 3.77 Å and 6.4 Å were reported, respectively. And the authors also have collectively

described a 3D-architecture of a viral milieu. The author has emphasized several times that they got the results from "cell extracts". Native cell extracts hold great promise for understanding the molecular structure of ordered biological systems at high resolution.

However, 1) the authors in fact have lysated the cells and isolated for several rounds, not in true native cell extracts. 2)As shown in the mass spectrum data (i. e. Figure 1e), the host cells are co-infected by L-A virus and L-BC virus. Since these two viruses are homologous. In addition, the virus L-A has different strains. Evidences are needed to prove that the newly found cation- π and flexible loops are not introduced by the L-BC virus or different strains. The authors should compare the structures of same L-A virus from "cell extract" and that purified from the cells using same infection way.3)The results from small number of sections can not support the results claimed in line 370-180. In fact, Fig.1C is quite different with Fig.6a. The particles's distribution and the milieu in Fig.1C and in Fig.6 are obviously different. "frequent observation" needs a statistic analyses.

Minor points:

- 1)Line 293: Fig. 5d is missing. Probably should be Supplementary Fig. 8b or 8c?
- 2)Line 488: The reference of the 'CryoSparc' should be cited when the first time it appears.
- 3)Line 489-495: This description here is not consistent with the Supplementary table 1. Please check it.
- 4)In Fig.4b and 4d, Asp540 should be marked so as to in accordance with the text.

LA Paper revisions

Reviewer #1 (Remarks to the Author):

The manuscript describes cell-extracted particle structures of yeast Sc-L-A virus. Only 3.4-Å X-ray crystal structure of the L-A virus was available. The significance of the study was to determine an atomic model of near native structure of cellular L-A virus particles. The most of so-far published structural studies of viruses focused on resolving a matured virus structure in extracellular phase. First, it is shown that mass spectrometric and morphological characterizations of the L-A virus particles and associating polysomes in yeast lysate fractions (Figs. 1 and 6, Suppl. Figs. 1, 10-15). Second, it is resolved a near-native structure of the matured yeast Sc-L-A virus particles at the resolution of 3.7-Å and discussed possible interactions and methyltransferase active site in the capsid (Figs. 2-4, Suppl. Figs. 2-6). Third, inner capsid structure is described at a low resolution (Fig. 5, Suppl. Figs. 7-9). Based on these findings, limited but new structural properties of the cellular L-A virus capsid seem to be revealed. However, I am skeptical to the significance and the purpose of studying “endogenous” structure of the L-A virus, and doubt an interpretation of the results, especially second and third findings.

Answer #1.0:

We appreciate the reviewer’s concise summary of our manuscript. To address their concerns, we collected additional EM data at an increased pixel size, which significantly improved the final resolution to 3.2 Å, and, therefore, propose the highest resolution structure of the mature L-A virus capsid reported to date. In addition, the images were acquired with our newly acquired and upgraded electron detector, Falcon IVi, improving the signal in the acquired micrographs, and therefore address the concerns of the reviewer by analyzing the high-resolution structure.

Main comments

1. Introduction P3 L54-75, P4 L108-L117, Discussion P11 L391-404

I doubt the Sc-L-A virus is an adequate target for fulfilling the significance since the Sc-L-A virus only infects intracellularly (endogenous), not extracellularly (exogenous), although I agree on the significance of resolving structure of the endogenous virus particles. The significance of the study is overinterpreted in “Introduction” and “Discussion” sections. Strictly speaking, a major contribution of the manuscript is to determine the first cryo-EM capsid structure and the atomic model of the matured L-A virus. The structural differences between former X-ray structure and the cryo-EM structure in this manuscript should not be due to the ones between endogenous and exogenous particles. The authors describe COVID-19 and other human pathogens as examples in Introduction and Discussion section, however they have both intracellular and extracellular phases and are enveloped ssRNA viruses, and thus they do not akin at all to the dsRNA non-enveloped yeast Sc-L-A virus with only intracellular phase. The same idea and methods cannot be applied for these intracellular/extracellular human viruses. I strongly recommend focusing on intracellular matured dsRNA L-A virus structure and reducing largely the description in Introduction and Discussion sections.

Answer #1.1:

We appreciate the reviewer's concerns, and we revised our manuscript accordingly. In response to the reviewer's suggestion, we have performed the following:

- (1) Previous **Introduction** on exogenous viruses [p. 3, "*This is exemplified by...multiple variants¹⁸*"] is now edited, and focus is placed on endogenous viruses and their roles, improving the corresponding part in the Introduction section by mentioning a specific example (human endogenous retrovirus K17) [p. 3 "*On the other hand, endogenous ... viral particle*"].
- (2) We have **improved the resolution** of the L-A virus cryo-EM structure, and we improved the impact of our major contribution by analyzing the highest resolution structure of the L-A virus to date (updated **Fig. 2** and **Fig. 3** and associated text accordingly). New text corresponding to this is the following:
 - Resolving the structure** [p. 6, "*To derive solid molecular... indicating critical structural rigidity*"] along with updating **Fig. 2**, **Fig. 3**, **Supp Fig 2** and **Supp Fig 4**.
 - Reanalysis of the cation- π interactions** [p.6-7, "*The cation- π interaction...folding and structural stability.*"] and reporting of their significance in the Discussion [p.10, "*We observed these...molecular level.*"] along with updating **Supp Fig 5** and adding **Supp Figs 6** and **7**.
 - Reanalysis of the reported loop regions**, [p.7, "*Structural changes compared ...for the virus's life cycle and function⁵³*"]
 - Discovery of a flexible loop in the active site** which is ordered in the crystal and flexible in the 3.2 Å cryo-EM map [p.8, "*Specifically, the flexible region...this binding pocket.*"] and reporting its significance in the Discussion section [p.10, "*An adjacent flexible loop...of the capsid*".]
- (3) Previous **Discussion** section on exogenous viruses [p.11, "*Fast and reliable screening ...the translation apparatus*"] is now replaced with a new paragraph on endogenous viruses [p. 10, "*Endogenous viruses have often...genetic and environmental factors⁶³.*"] and focus is placed on endogenous viruses and the impact of our work in their structural study.

2. Results P6 L180 – P7 L228, Fig. 2, Suppl. Fig. 3-4, Discussion P11 L413 – L418

I am not convinced the description with regards to side chain interactions and conformations at the resolution of 3.7-Å. The achieved resolution of the cryo-EM model (3.7-Å) is worse than available crystal structure (3.4-Å). In this resolution, it is very difficult to interpret the side chain interactions such as cation- π stacking and side chain conformational variations. The atomic model is largely biased by the achieved resolution of the cryo-EM map and the side chains can be overfitting to the map (This can also be a reason of cc value is unexpectedly high for 3.7-Å cryo-EM map in suppl. fig. 4.). As shown in Fig. 2, the side chains look like a bump and it is difficult to determine the accurate orientations. However, it might be possible to compare the experimentally obtained cryo-EM map (3.7-Å) with a calculated density map from the crystallographic atomic model at a 3.7-Å. It should be justified the descriptions by considering the achieved resolution.

Answer #1.2:

We thank the reviewer for their comment. We eventually resolved the structure of the L-A virus at higher resolution to address their concern, please see detailed **Answer**

#1.1. This structure is considerably superior in terms of resolution to all prior work on the L-A virus structure. This now enables us to assign side chain densities (updated **Supp Table 1**, updated **Fig. 2**), and we are highly confident that our findings are now robust and reliable which are reported in **Answer #1.1**.

3. Results P7 L230 – P8 L251, Discussion P12 L420 – L432

It is not easy for me to see the differences of the flexible loops between X-ray and cryo-EM structures in Fig. 4 and seems to be quite limited differences. It looks both structures show an opened conformation. For example, in main text (P7 L245), the authors mention, “their distance is 3-Å closer in the cryo-EM structure”, however the achieved resolution is 3.7-Å, which cannot interpret the structural difference of the 3 Å. It is too speculative that these additional flexible loops contribute to open and closed states of capsid methyl transferase active site. Again, X-ray and cryo-EM structures should not reflect the differences of exogenous and endogenous structures. I do not think the cryo-EM structure shows more functional conformation except crystal packing.

Answer #1.3:

We thank the reviewer for their comment. We have now re-analyzed the loop regions of the L-A virus capsid at 3.2 Å resolution where model building in maps of such quality is mostly guided by map density. We have revised our model because we identified a loop in the active site which is highly flexible (exhibits low local resolution) while in the crystal it is resolved. We attribute this difference in its possible regulatory role. We have included this finding in updated figure panels **Fig. 3c** and **Fig. 3f**, and describe it in a new Results text [*p. 8, “Specifically, the flexible...binding pocket”*].

4. Results P8 L265 – P9 L316, Fig. 5

The analysis of this part has not been performed well technically and is not reliable. I strongly suggest removing this part fully from the manuscript. It is not described the details at all how to reconstruct the interior structure in Methods section (C1 reconstruction). The 2D-class averaged images after subtracting capsid intensity contribution are very strange (Fig. 5A). The intensity of the capsid is still largely remained after the subtraction. In Fig. 5B and 5C, why is only a portion of the pol/genome density revealed? The viral genome density should be distributed more evenly in the capsid. The obtained map might be pol or viral genome or something else at the low resolution. The AlphaFold2 prediction does not help for determining the position of the capsid and the pol since there is no reliability of positioning the capsid and pol as clearly shown in results in suppl. fig.8b (The linker region is poorly predicted in green region, left figure. No reliability of positioning capsid and pol shown in right figure).

Answer #1.4:

We followed the reviewer’s comment, and this section along with the supporting computations was replaced with the major focus on the L-A virus structure - the focus of the paper has shifted to the revised structure of the L-A virus at 3.2 Å (see **Answer #1.1**). Due to this shift of focus, we have included more solid results that are explained in detail in **Answer #1.1**.

Minor comments

Introduction

P3 L54-55 How does the structural analysis of viruses contribute to epidemiology and ecology? It may be, but it is a confusing description.

Answer #1.5:

Section has been re-written to clarify the comment of the reviewer [p. 3, *The structural analysis...viruses in cells*].

P3 L55-56 Virus diagnostics (virus detections in cells) were traditionally performed serologically and genetically. Or, the authors means morphological characterizations of viruses?

Answer #1.6:

Section has been re-written to clarify the comment of the reviewer [p. 3, *The structural analysis...viruses in cells*].

P3 L72-75 I do not understand the intended meaning. Please clarify.

Answer #1.7:

We have revised and refined the sentence for enhanced comprehension [p. 3, *On the other hand...the viral particle*]. The statement addresses potential concerns associated with the (over-)expression of viruses in non-native environments, such as cell culture lines rather than living organisms, or overexpression using truncated promoters. Such alterations may result in structural changes unique to the expression system, thereby increasing the likelihood of structural artifacts.

P4 L97 Double-stranded ds(RNA)

Answer #1.8:

The sentence now reads as “Double-stranded RNA (dsRNA) viruses ...”.

P4 L109 “after successful biochemical enrichment”. This description is clearly too much. What author did was to lyse infected yeast cells and to purify virus fractions using conventional size exclusion column purification, not biochemical enrichment. It may mean MASS analysis, but then please say so.

Answer #1.9:

We have replaced “after successful biochemical enrichment” with “identified within fractionated yeast cell extracts”.

Results

P5 L121 “cryo-EM accessible” Cryo-EM can be accessible to varieties of specimens including cellular samples. It is a confusing description.

Answer #1.9:

We have changed the title to “*Identification of the L-A virus in eukaryotic cell extracts through cryo-EM*”.

P5 L134, Fig. 1c How is the complex confirmed to be a FAS in a raw cryo-EM image?

Answer #1.10:

Our team has extensive experience in analyzing Fatty Acid Synthase with cryo-EM, as demonstrated in prior studies (e.g., Tüting et al. *Proteomics* 2023; Skalidis et al., *Structure*, 2022; Kastritis et al, *MSB*, 2017), and the 2D signature of FAS in micrographs is distinct and easily identifiable. We used the original micrographs in another research project, which focused on automatically identifying medium resolution cryo-EM map densities, taking FAS as a prime example. The following are the results of the FAS model fitted into the medium-resolution derived density [Tüting et al. *Proteomics*, 2023, doi: 10.1002/pmic.202200096].

P5 L159-160 I do not think that enriching the L-A virus is unique to this study compared with other studies. The beads beating cell lysis and fractionations using SEC is a regular method for many other studies.

Answer #1.11:

It is correct that L-A virus was previously enriched by other methods, while cell lysis and SEC fractionation is a regular method for studying cellular material. However, applying this method to the L-A virus is novel. Our approach is distinct in that we used a simple and efficient setup (lysis, concentration, and size-exclusion) to mildly reduce sample complexity while preserving endogenous interactions within rapidly retrieved yeast cell extracts directly amenable for cryo-EM analysis. This contrasts with classical L-A virus preparations, which involve PEG precipitation and eliminate all other native interactions, retrieving only intact capsids. We have revised the text to clarify this point and emphasize the uniqueness of our methodology [p. 6, “*Compared to other...quantitative mass spectrometry*”].

P6 L173 I understand that it has often describe to be T=2 in some icosahedral dsRNA viruses since they have two identical subunits in an asymmetric unit. However, T=1 should be a correct description. T=2 icosahedron does not mathematically existed according to the definition of T number ($h^2 + hk + k^2$).

Answer #1.12:

We appreciate the reviewer’s attention to the triangulation number issue. The L-A virus is indeed composed of 120 Gag monomers organized into 60 dimers, resulting in a triangulation number that does not adhere to the Caspar and Klug principle. Although this leads to a forbidden triangulation number of T=2, the L-A virus is often imprecisely classified as T=1. We have revised the text to accurately describe the unique characteristics of the L-A virus and its deviation from the Caspar and Klug principle. We have also included an appropriate citation to “*Icosahedral Symmetry of Viral Capsids*” by Anze Lošdorfer Bozic to support this information [p.6, “*The icosahedral protein shell...crystallographic counterpart (Supplementary Fig. 4)*”].

P7 L226-228, Suppl. Fig. 5b I may not fully understand the analysis, however if A-B dimer is formed initially, all capsid protein form the A-B dimer for assembling? If so, then monomer subunits do not exist a lot? I am not really believed that monomer units are one by one gathered sequentially to make a capsid.

Answer #1.13:

We would like to clarify to the reviewer that this is not a model of virus assembly – it is a visual representation of the stable interfaces according to calculated energetics of all possible interfaces within the capsid. During assembly, highly complex interactions occur that are non-additive and go beyond binary interfaces. We agree that it is unlikely for monomeric Gag to not dimerize before capsid association, especially considering the close connection between polysomal assemblies and non-mature viral particles (Supp Fig. 10; Supp. Fig. 11); for possible assembly mechanisms of L-A/L-BC virus we refer the reviewer to our determined immature capsids that are similar to those reported for the L-BC virus [<https://www.nature.com/articles/s42003-022-03793-z>]. We included these clarifications in the Results section [p. 8, “*This analysis does not... clusters of monomers instead*⁵⁵.”]

P8 L252-264, Suppl. Fig. 6 There is no description of DALI score in the main text and figure legend. Are they sufficiently reliable to say distant structural homologs? According to a DALI developer’s instruction, a Z-score between 2 and 8 is a grey area. It is not convincing to say “as well as structural homologs with potential evolutionary implications” in L263-264. It is still very speculative and very challenging to find its evolutionary relation to other methyl transferases.

Answer #1.14:

We appreciate the reviewers’ comment, and therefore, all analysis is now focused on the L-A virus structure, replacing this analysis (see **Answer #1.1**).

P10 L336-344 I am not an expert of ribosome structures, however I feel it is needed to show structural differences of the 3D reconstructions of the active ribosomes. In the intermediate resolution (8-10 Å) in suppl. fig.11c, is it possible to distinguish their different active states including tRNA occupancy?

Answer #1.14:

We would like to thank the reviewer for their comment – we do not think we need to show differences across active states as this goes beyond the scope of our manuscript. In our view, bound tRNAs are enough evidence of an active translational state.

P11 L376 – L379, Fig. 6b How was a segmentation of the image done for viruses and ribosomes/interacting proteins in the entire cell image (Fig. 6b)?

Answer #1.15:

We thank the reviewer for this inquiry about the segmentation of the image for viruses and ribosomes/interacting proteins in the entire cell image (Fig. 6b). We apologize if the details were not clear in the manuscript. The process of obtaining these images

involved high-pressure freezing, cryo-substitution, resin embedding, and sectioning with an ultramicrotome, as described in the Methods section under "High-pressure freezing." In brief, yeast cells were rapidly frozen, cryo-substituted, and embedded in HM20 resin. Ultrathin sections (30 nm) were prepared using an ultramicrotome, transferred to formvar-coated copper grids, and poststained with uranyl acetate and lead citrate. The samples were then observed with a transmission electron microscope operating at 80 kV. To create the schematic representation shown in Fig. 6b (right panel), we manually traced the original micrograph using Adobe Illustrator with a graphics tablet. This allowed us to accurately outline the viruses and ribosomes/interacting proteins based on the observed features.

Methods

P13 L477 – 479 Any energy filter was used for improving obtained images of thick virus samples?

Answer #1.16:

We did not use an energy filter for improving the obtained images of thick virus samples in this study as our Glacios microscope does not accommodate one. However, the new acquisition includes an updated electron detector (Falcon IVi) that improves the signal-to-noise ratio, leading to higher resolution of the L-A virus structure.

P13 L483 – P14 L510 It is straightforward and sounds better to put accurate particle numbers in this section, e.g. "approximately 230000 particles" should be the exact number.

Answer #1.17:

The reviewer is right, exact numbers would sound better. But as this number does not add any valuable content to our results, we completely omitted this sentence. Overall, exact numbers that were used for the 3D reconstructions are mentioned in Supplementary Table 1.

Reviewer #2 (Remarks to the Author):

In this manuscript Schmidt et al., used single-particle cryo-EM to visualize the yeast L-A endogenous virus in near native environment. Major finding of this study is the ability to use cryo-EM to visualize the virus in its native state in a cell and show interaction with cellular machinery (ribosomes). It is surely a technological advancement and will be of interest to wider field.

Authors were able to determine the structure of L-A capsid to a modest resolution of 3.77Å and build a c-alpha model. Even though the crystal structure of capsid has been previously solved, authors found some novel interactions and dynamic flexibility in their cryo-EM structure.

Even though the technological advancement is obvious, it remains to be seen if this approach can identify a novel protein correctly. Most of the biological findings in the study are not novel and have been reported earlier that viral replication involves concentration of cellular machinery named "factories".

Authors should speculate the feasibility of this approach to determine the structure of a unknown protein interacting with viral components if the X-ray structure is not available for unknown protein.

Answer #2.1:

We sincerely appreciate the reviewers thoughtful summary and the highlighting of the novel aspects of our work. The reviewer comments indeed point out important areas that we have addressed during our revision.

We are pleased to inform that, upon revising, we managed to significantly enhance the resolution of the viral capsid to 3.21 Å via a new acquisition at a higher pixel size. With this enhanced resolution, we are now more confident in validating our previous findings, such as the dynamic loop flexibility and the abundance of cation- π interactions.

Regarding the question on the identification of a novel protein, we want to assert that our approach is indeed capable of accurately identifying such proteins. Initially, we utilized our map for manual backbone tracing, and used the DALI webserver for homologous signature search. In the revised manuscript, we deployed the ModelAngelo automatic model building toolkit, without supplying any sequence, to model the protein in our density [p. 5, "Simultaneously, we used ModelAngelo46...(Supplementary Fig. 3)."]. ModelAngelo constructed a somewhat fragmented atomic model; We used the predicted sequences of these structural fragments to conduct a BLAST search against the non-redundant (nr) database. Remarkably, the top hit was the L-A virus [p. 5, "Simultaneously, we used ModelAngelo46...(Supplementary Fig. 3)."]. These results are summed up in the concluding sentences of the section [p. 6, "Compared to other studies...quantitative mass spectrometry"]

So, we believe that if the resolution is sufficiently high to provide at least partial side chain density, AI-powered tools such as ModelAngelo can indeed identify a protein accurately. For instances of lower resolution, we would like to refer to our recent publications in Structure (Skalidis et al., 2022) and Proteomics (Tüting et al., 2023), where we developed unsupervised identification workflows to recognize biomolecules in lower resolution cryo-EM maps from cell extracts.

Additional comments-

Page 5- line 130 and Page 13, line 443. Please elaborate the washing of cell pellet with water. Was it resuspended in water and spun or just rinsed in water?

Answer #2.2:

We corrected the Methods section to “The pellet was washed by resuspending in distilled water and recentrifuged with the same parameters (resulting pellet ~7 g).”

Page 5, line 157- Do authors mean there were 1.8 \pm 0.49 polymerase present per capsid? That would be significant more than 60:1 ratio of gag to gag-pol ratio. Ref 31 had suggested 1.9% of ribosomes producing gag-pol after -1 frameshifting.

Answer #2.3:

We apologize for any confusion caused. To clarify, 2 polymerases per capsid means 2 pol per 120 gag, which is in-line with the suggested ratio. But after discussions with the co-authors, we decided not to include this data in the revised version as stoichiometry calculations from quantitative MS data are highly challenging for the L-A virus as peptides from the polymerase are not abundant. Therefore, we decided to omit this part from the manuscript for the sake of clarity and accuracy and kept the MS data only for identification of the proteomic content (please see Fig. 1e, Supplementary Figs. 4).

What is the biological significance of cation- π interactions identified in this study (Fig 2). It needs to be confirmed by site-directed mutagenesis.

Answer #2.4:

A cation- π interaction is a unique form of intra-protein interaction that can only occur when both residues are in the correct distance and conformation, with the cationic moiety oriented perpendicularly to the delocalized π -electron system. This specific configuration implies that the cation- π interaction is only feasible in correctly folded proteins, contrasting with other intra-protein interactions such as hydrogen bonding, ionic interactions, desolvation, and van der Waals interactions. The exceptional resolution of the residues involved in this interaction further supports this notion.

Regarding the suggestion of using site-directed mutagenesis to elucidate the role of this interaction, we acknowledge its potential usefulness. The virus under investigation is present in an unaltered ATCC wild-type strain. Expressing a mutated viral capsid would necessitate isolating the genomic information, acquiring a "clean" yeast strain, and also dealing with the associated killer virus, which is yet unknown in our case.

However, we must highlight that implementing this approach is beyond the scope of our current study – we have instead validated their presence by resolving the higher resolution structure of the L-A virus by cryo-EM at 3.2 Å and the interactions are persistent. In addition, we have performed a large-scale analysis of their presence in *Duplornaviricota* and found they exist across all members of the phylum with existing capsid structures [p. 6, “The cation- π interaction represents...folding and structural stability.”].

Page 9- line 293 describes Fig 5d but I could not find Fig 5D in figure.

Answer #2.5:

We thank the reviewer for pointing this out. During revisions, the entire section including Figure 5, was omitted after suggestions from Reviewer #1.

How were the ribosome particles picked (manual, blob, template or combination thereof ?) during the cryo-EM processing (Supp Fig 11). Its description was missing from methods. It will help wider cryo-EM community to pick particles of similar dimension containing different viral (capsid here) or cellular macromolecule targets.

Answer #2.6:

We revised the Methods section, now stating that the ribosomal analysis was blob picked.

Page 10- line 348-349. How do authors consider the fact that partially assembled viral particles (stage II-VI in Fig 1a) will not be as electron dense as the whole particle and that will make picking them from micrographs unreliable. Unless the viral replication is synchronized by external factors there should be broad heterogeneity in viral particles and majority may not be in mature state.

Answer #2.7:

This is an excellent remark. We also thought about this during analysis, which is the reason, we did not rely on automatic picking of partially ensembled viral particles. We added the following sentence to the figure legend (Supplementary Figure 10):

“Statistics of (d) and (e) are derived from manual picking, due to lower contrast of partially assembled particles.”

Page 14- line 504, Coot is not a manual refinement program. Appropriate word may be “interactive refinement”.

Answer #2.8:

We changed the sentence - It’s now reading as:

“subjected to iterative refinement cycles using Coot⁷⁰ and PHENIX⁷³ yield the final coordinates.”

Reviewer #3 (Remarks to the Author):

In this work, an icosahedral reconstruction and an asymmetrical reconstruction of the L-A virus retrieved from cell extracts at 3.77 Å and 6.4 Å were reported, respectively. And the authors also have collectively described a 3D-architecture of a viral milieu. The author has emphasized several times that they got the results from "cell extracts". Native cell extracts hold great promise for understanding the molecular structure of ordered biological systems at high resolution.

Answer #3.1:

We are grateful for the reviewer's acknowledgment of the value of using native cell extracts in our work. The utilization of these extracts is indeed pivotal in facilitating a comprehensive understanding of the molecular structure of ordered biological systems at high resolution. We believe that our work with the icosahedral and asymmetrical reconstructions of the L-A virus, retrieved directly from these cell extracts, has illustrated this advantage and contributed to the broader field of structural biology. We would like to note that during revisions we improved the resolution of the L-A virus and reached 3.2 Å, and therefore, improved the interpretation of the cryo-EM map with a more solid model.

However,

1) the authors in fact have lysated the cells and isolated for several rounds, not in true native cell extracts.

Answer #3.2:

We thank the reviewer for their comment: In our study, the extracts were produced from unaltered cells through a gentle fractionation process based on Stokes radii, without any harsh treatment, preserving their native state as much as possible. As such, we describe our extracts as being in a "near-native" state. Indeed, the advances in cryo-electron tomography combined with cryo-focused ion beam scanning electron microscopy (cryo-FIB-SEM) have dramatically reduced the disruption of cellular interactions, providing an avenue for analysis of vitrified cell slices. However, with current technologies, "true" native analysis (only lysate without fractionation) may still face challenges due to smaller proteins and other biomolecules that can obstruct the view in cryo-EM, e.g., ribosomes. Regarding this point, we performed a single round of size-exclusion chromatography (SEC) and analyzed the resulting fractions. We do not consider this as "several rounds" of isolation.

2) As shown in the mass spectrum data (i. e. Figure 1e), the host cells are co-infected by L-A virus and L-BC virus. Since these two viruses are homologous.

Answer #3.3:

Indeed, the co-infection of yeast cells with L-A and L-BC viruses is common, and as the reviewer correctly pointed out, it is not surprising to observe this in our mass spectrometry data. Homologous L-BC virus typically exists in lower copy numbers, approximately 10 to 20% compared to L-A virus. To provide clarity and further support for our findings, we have added the following reference to our manuscript citing this part [p.4, "*The L-BC virus is...the satellite viruses*²⁵"] [reference: Rodríguez-Cousiño, N., & Esteban, R. (2017). Applied and Environmental Microbiology, 83(4), e02991-16. <https://doi.org/10.1128/AEM.02991-16>]. This study discusses the relationships and evolution of these double-stranded RNA Totiviruses in yeast populations, and reinforces our observations on the co-infection of L-A and L-BC viruses. We thank the

reviewer for this keen observation and the opportunity to enhance the robustness of our work.

In addition, the virus L-A has different strains. Evidences are needed to prove that the newly found cation- π and flexible loops are not introduced by the L-BC virus or different strains.

Answer #3.4: We appreciate the reviewer's suggestion, and it has indeed prompted us to carry out additional validation. Regarding the potential influence of different L-A virus strains, we performed a comprehensive comparison of all UniProt entries that include "*Saccharomyces cerevisiae* virus L-A". This included a total of 17 strains such as "L-A", "L-A-2", "L-A-lus", and "L-A-28". These were all aligned to the sequence used in our analysis (UniProt Identification: P32503).

The residues involved in the cation- π interactions we observed are highly conserved (100%), except for Phe561, which is mutated to Tyrosine in 15 sequences (it is still an aromatic residue, though, retaining the cation- π interaction). Similarly, the flexible loop regions exhibit high sequence conservation, with only a few exceptions, listed in the following table:

Strain variants of the flexible regions. Only residues with a sequence identity less than 100% are listed.

Residue	Identity	Amino acid	Mutations
94	0.125	R	['R', 'R', 'K', 'K', 'K', 'K', 'K', 'K', 'K', 'K', 'K', 'K', 'K', 'K', 'K', 'K', 'K', 'K']
105	0.375	A	['A', 'A', 'T', 'T', 'T', 'A', 'A', 'T', 'T', 'T', 'T', 'A', 'A', 'T', 'T', 'T']
527	0.25	V	['V', 'V', 'V', 'V', 'A', 'A', 'A', 'A', 'A', 'A', 'A', 'A', 'A', 'A', 'A', 'A', 'A', 'A']
528	0.0625	Y	['H', 'Y', 'Q', 'Q', 'C', 'C', 'C', 'C', 'C', 'C', 'C', 'C', 'C', 'C', 'C', 'C', 'C', 'C', 'C', 'F', 'F']
530	0.375	D	['D', 'D', 'D', 'D', 'N', 'N', 'N', 'N', 'N', 'N', 'N', 'N', 'N', 'N', 'N', 'N', 'N', 'D', 'D', 'D']
531	0.75	T	['T', 'T', 'V', 'V', 'T', 'T', 'T', 'T', 'T', 'T', 'T', 'T', 'T', 'T', 'T', 'T', 'T', 'V', 'V']
535	0.375	T	['T', 'T', 'T', 'T', 'S', 'S', 'S', 'S', 'S', 'S', 'S', 'S', 'S', 'S', 'S', 'S', 'S', 'S', 'T', 'T']
602	0.125	A	['A', 'A', 'T', 'T', 'T', 'T', 'T', 'T', 'T', 'T', 'T', 'T', 'T', 'T', 'T', 'T', 'S', 'S']
603	0.125	H	['H', 'H', 'F', 'F', 'F', 'F', 'F', 'F', 'F', 'F', 'F', 'F', 'F', 'F', 'F', 'F', 'F', 'F', 'F', 'F', 'F']
604	0.0625	A	['S', 'A', 'S', 'S', 'S', 'S', 'S', 'S', 'S', 'S', 'S', 'S', 'S', 'S', 'S', 'S', 'S', 'S', 'S', 'S', 'S']
609	0.9375	S	['R', 'S']

None of the listed residues is to be expected to change the overall structure, as in most of the mutations, the overall properties of the residue (size, charge, polarity, etc.) are preserved.

As for the potential influence of the L-BC virus on our reconstruction, we believe it is unlikely to have an impact. Based on our mass spectrometry data, the L-BC virus constitutes about 10% of the L-A virus abundance. This is in-line with previous results, showing a co-infection of L-BC with the L-A helper system. Even if both viral particles were included in the cryo-EM reconstruction, the influence of the L-BC virus would be negligible. If anything, L-BC particles would decrease the resolution, not increase it as we have observed in the case of cation- π -stabilized residues. Also, the L-BC capsid is approximately 6 Å larger than that of the L-A virus. (Figure R2) This difference would allow for the separation of their signatures during image processing.

Figure R2: Backbone atom distance distribution to the center of mass (CoM) of the respective viral capsid.

Furthermore, to eliminate any possible impact on the cation- π networks identified in the L-A virus, we superimposed the structures of both viral capsids. We then selected all corresponding side chain densities in the L-BC virus within a 3 Å radius around the cationic and π -electron network. We found that none of the identified residues have functional counterparts in the L-BC virus. This gives us high confidence that the results we presented are not artificially assumed due to sample contamination.

The authors should compare the structures of same L-A virus from "cell extract" and that purified from the cells using same infection way.

Answer #3.5:

We appreciate the reviewer's suggestion to compare the structures of the L-A virus obtained with different methods. The L-A virus we worked with is endogenous to the ATCC wild-type strain used in this study. This virus is not a result of infection; instead, it is inherently present in the cells. This is also the case in many other scientific strains as well as industrial strains, including those used in winemaking. Our protocol for purifying the virus is both mild and swift. This allows us to maintain native assemblies as observed in the virus:ribosomal assemblies. Furthermore, we have already compared our structure with the previously published structure (1m1c), which was purified from yeast cells to homogeneity and in addition, during revisions, determined the structure of the L-A virus at 3.2 Å, the highest reported to date for this virus. See Figure 2, in which side chain densities are now fully recapitulated. We hope this clarification addresses the reviewers concerns and assures that our methodology is well-aligned with the study's objectives and requirements.

3)The results from small number of sections can not support the results claimed in line 370-180. In fact, Fig.1C is quite different with Fig.6a. The particles's distribution and the milieu in Fig.1C and in Fig.6 are obviously different. "frequent observation" needs a statistic analyses.

Answer #3.6:

We acknowledge the reviewer's observation regarding the differences between Fig.1c and Fig. 6a. These differences stem from the distinct emphases of the two figures. In Fig. 1c, our objective was to depict the complexity and heterogeneity of the sample, hence we chose a typical image that contained viral particles with binding partners, FAS, and other macromolecules. On the other hand, in Fig.6, the focus was strictly on viral assemblies.

However, we agree with this suggestion that our observations require additional statistical validation. The term "frequent observation" was used to describe the recurring phenomena we encountered during our research, but it is clear that this claim would be strengthened by a more rigorous statistical analysis. We exchanged the term "frequent observation" to "common observation", to clearly indicate, that this is a subjective term [p.9: "*A common observation in the cryo-EM micrographs*]. We greatly appreciate the insightful comment and revised our manuscript accordingly to ensure that our results and conclusions are accurately supported by the evidence presented.

Minor points:

1)Line 293: Fig. 5d is missing. Probably should be Supplementary Fig. 8b or 8c?

Answer #3.7:

During revision, Figure 5 and the respective paragraph describing it was omitted after comments from reviewer 1.

2)Line 488: The reference of the 'CryoSparc' should be cited when the first time it appears.

Answer #3.8:

We added the appropriate reference here. Also, we fixed the spelling to the correct form: cryoSPARC, as by the title of the original publication: "*Punjani, A., Rubinstein, J., Fleet, D. et al. cryoSPARC: algorithms for rapid unsupervised cryo-EM structure determination. Nat Methods 14, 290–296 (2017). <https://doi.org/10.1038/nmeth.4169>*"

3)Line 489-495: This description here is not consistent with the Supplementary table 1. Please check it.

Answer #3.9:

We revised the method section and eliminated any inconsistencies between the written Method section and the Supplementary Figure 1, containing the microscope and refinement parameters. In the method sections (p. 9), we now reported the correct numbers ("After removing low-resolution data, final datasets of 17000 particles (L-A virus; Falcon 3; #1), 17411 particles (Ribosomes; #2), 17007 particles (L-A virus; Falcon 3; #3), and 12974 particles (L-A virus; Falcon 4; #4) were used for final 3D reconstruction."), which are coherent with the data reported in table S1 ("Final particle images (no.) 17000 17411 17007 12974").

4)In Fig.4b and 4d, Asp540 should be marked so as to in accordance with the text.

Answer #3.10:

During revision, the original figures 3 and 4 were replaced with new ones. The analysis of the binding pocket is now found in Figure 3. On page 7, we refer to this figure in text [p.7: *specifically in loop regions (R94–V104; Y528–L534, G601–H607) of protomers A and B, respectively (Fig. 3a-d).*], and in-text mentioning and the figure are in accordance.

REVIEWERS' COMMENTS:

Reviewer #1 (Remarks to the Author):

The authors have determined the 3.2 Å cryo-EM structure of the yeast Sc-L-A virus, extracted from enriched fractions of yeast cell extracts. The cryo-EM structure of the viral capsid has been intensively compared with the previously reported X-ray crystal structure. The interfaces of the subunits within the capsid have been comprehensively assessed among dsRNA viruses of the same evolutionary lineage. Notably, the findings highlight the potential contributions of cation- π interactions to the stability of icosahedral dsRNA viruses, a feature previously not well described. The possible cellular interactions between viruses and polysomes emphasize the importance of studying intracellular virus particles.

The authors have significantly improved the manuscript by carefully addressing almost all of my previous comments. The revised manuscript clearly indicates the significance of studying intracellular particles. Although the improved cryo-EM model is not considered high-resolution from my perspective, it is adequate for comparison with X-ray crystal structures and for discussions regarding subunit interfaces. Some minor comments still need to be addressed.

Minor Comments:

Line 183: The use of a triangulation number $T=2$ should be avoided. According to the Caspar and Klug principle, $T=2$ does not exist. Totiviridae viruses adopt a $T=1$ organization, consisting of 60 structural units, with each unit comprising two chemically identical capsid proteins, thus totaling 120 protomers.

Lines 191-197, Lines 203-211, Supplementary Figures 6-8: The discussions on structural characterizations and cation- π interactions across other dsRNA viruses are crucial for understanding their capsid stability and assembly. I recommend incorporating these discussions into the main figures, if possible.

Lines 340, 341: AIDS is the only disease listed here, whereas Ebola and SARS-CoV1/2 are names of viruses, not diseases or symptoms. This distinction should be clarified.

Figure 2: In the figure legend, I am unable to identify a yellow star in panel 2b. Additionally, it should show cryo-EM map of 3.21 Å to verify fitting of atomic model in panel 2c.

Reviewer #2 (Remarks to the Author):

Authors have adequately addressed my concerns.

Reviewer #3 (Remarks to the Author):

I am generally satisfied with the revision and authors' responses to me.

LA Paper revisions

Reviewer #1 (Remarks to the Author):

The authors have determined the 3.2 Å cryo-EM structure of the yeast Sc-L-A virus, extracted from enriched fractions of yeast cell extracts. The cryo-EM structure of the viral capsid has been intensively compared with the previously reported X-ray crystal structure. The interfaces of the subunits within the capsid have been comprehensively assessed among dsRNA viruses of the same evolutionary lineage. Notably, the findings highlight the potential contributions of cation- π interactions to the stability of icosahedral dsRNA viruses, a feature previously not well described. The possible cellular interactions between viruses and polysomes emphasize the importance of studying intracellular virus particles.

The authors have significantly improved the manuscript by carefully addressing almost all of my previous comments. The revised manuscript clearly indicates the significance of studying intracellular particles. Although the improved cryo-EM model is not considered high-resolution from my perspective, it is adequate for comparison with X-ray crystal structures and for discussions regarding subunit interfaces. Some minor comments still need to be addressed.

Answer #1.0:

We thank the reviewer for the positive review of our revised manuscript. We will address all remaining minor points.

Minor Comments:

Line 183: The use of a triangulation number $T=2$ should be avoided. According to the Caspar and Klug principle, $T=2$ does not exist. Totiviridae viruses adopt a $T=1$ organization, consisting of 60 structural units, with each unit comprising two chemically identical capsid proteins, thus totaling 120 protomers.

Answer #1.1:

We removed the sentence from the manuscript to:

The icosahedral protein shell has a triangulation number $T=1$ organization, consisting of 60 structural units, with each unit comprising two chemically identical capsid proteins, formed by 120 protomers, with overall diameter (400 Å) and thickness (46 Å) comparable to its crystallographic counterpart (**Supplementary Fig. 5a**).

Lines 191-197, Lines 203-211, Supplementary Figures 6-8: The discussions on structural characterizations and cation- π interactions across other dsRNA viruses are crucial for understanding their capsid stability and assembly. I recommend incorporating these discussions into the main figures, if possible.

Answer #1.2:

We thank the reviewer for this feedback on the cation- π interactions. We would like to keep the current organization as it is. We also believe that these specific type of interaction is very important for the stability due it's unique configuration. But for example the structural comparison (Supp. Figure 8) is very limited by the number of available capsid structures, we think that the manuscript would not benefit of integrating these data into a main figure.

Lines 340, 341: AIDS is the only disease listed here, whereas Ebola and SARS-CoV1/2 are names of viruses, not diseases or symptoms. This distinction should be clarified.

Answer #1.3:

We changed AIDS to HIV, as we were talking about viruses and their implication, not mentioning the resulting diseases.

Figure 2: In the figure legend, I an unable to identify a yellow star in panel 2b. Additionally, it should show cryo-EM map of 3.21 Å to verify fitting of atomic model in panel 2c.

Answer #1.4:

We thank the reviewer for this comment. In fact, the figure legend is wrong, as the figure changed during the revision process. Initially, a side-by-side comparison of both maps were planned, but dropped in favor of the high resolution map. We harmonized both, figure and figure legend to the 3.21 Å map.

Reviewer #2 (Remarks to the Author):

Authors have adequately addressed my concerns.

Reviewer #3 (Remarks to the Author):

I am generally satisfied with the revision and authors' responses to me.

Answer #2.0:

We thank both reviewers for the positive feedback to our revised manuscript.